# ImmunoMatch learns and predicts cognate pairing of heavy and light immunoglobulin chains

Dongjun Guo ®[1,2,3], Deborah K. Dunn-Walters[4], Franca Fraternali ®[1,2,5] ✉ & Joseph C. F. Ng ®[1,2,5] ✉

The development of stable antibodies formed by compatible heavy (H) and light (L) chain pairs is crucial in both in vivo maturation of antibody-producing cells and ex vivo designs of therapeutic antibodies. We present ImmunoMatch, a machine-learning framework trained on paired H and L sequences from human B cells to identify molecular features underlying chain compatibility. ImmunoMatch distinguishes cognate from random H–L pairs and captures differences associated with $\kappa$ and $\lambda$ light chains, reflecting B cell selection mechanisms in the bone marrow. We apply ImmunoMatch to reconstruct paired antibodies from spatial VDJ sequencing data and study the refinement of H–L pairing across B cell maturation stages in health and disease. We find further that ImmunoMatch is sensitive to sequence differences at the H–L interface. These insights provide a computational lens into the broader biological principles governing antibody assembly and stability.

The immune system produces an extraordinarily diverse repertoire of antibodies to combat a wide range of immune challenges from invading pathogens as well as endogenous, aberrantly expressed antigens in contexts such as cancer. Our antibody repertoire is diverse, encompassing over $10^{12}$ distinct antibodies[1–4]. Such diversity is achieved mainly by two processes: first, a random recombination of genetic fragments in the immunoglobulin gene locus occurs independently for the two types of antibody chains, the heavy (H) and light (L) chains[5–9], which assemble to form an antibody molecule. These recombination processes already generate $2 \times 10^6$ different combinations of H–L chains[10], the sequence diversity of which is further potentiated by the imprecise joining of these gene fragments[11–14]. Second, mutations accumulated in the antibody variable region substantially increase repertoire diversity[15–18]. Investigation into the nature, dynamics and regulation of the antibody response in vivo[19–21], as well as the engineering of highly specific antibodies[22–24], are both active fields of modern biomedical research.

Studies of antibodies have gradually recognized the importance of a wide variety of 'developability' factors beyond the binding affinity of the antibody to its antigen[25–28]. One such issue pertains to thermostability, which is crucial in ensuring that the H and L chain can assemble to constitute a manufacturable, functional antibody therapeutic[29–31]. Using high-throughput sequencing methods, to identify stable H–L pairs, researchers generated separate sequencing libraries of H and L chains, manually paired expanded H and L clonotypes, and expressed and tested these antibodies in vitro[32,33]; this has been, for instance, applied to identify broadly neutralizing antibodies against HIV-1 (refs. 34,35). Recent single-cell methods produce paired H–L sequences from thousands of antibody-producing cells[36–39], which can circumvent problems in the traditional approaches, yet these methods are costly and still fall short of the true repertoire diversity[40]. The study of how stable antibody H–L pairs are generated is also relevant in basic biology: B cells, precursors to the antibody-secreting plasma cells, express B cell

[1]Research Department of Structural and Molecular Biology, Division of Biosciences, University College London, London, UK. [2]Institute of Structural and Molecular Biology, University College London, London, UK. [3]Randall Centre for Cell & Molecular Biophysics, King's College London, London, UK. [4]School of Biosciences and Medicine, University of Surrey, Guildford, UK. [5]Department of Biological Sciences, Birkbeck, University of London, London, UK. ✉e-mail: f.fraternali@ucl.ac.uk; joseph.ng@ucl.ac.uk

receptors (BCRs) comprising mainly the membrane-bounded version of antibody molecules[41,42]. During its development in the bone marrow, a B cell undergoes checks to ensure its BCR can be stably assembled and thus can sustain cellular signals to maintain viability[43–46], while removing autoreactive B cells via cell death[47–50]. This poses an inherent challenge in characterizing cases where stable, non-autoreactive H–L pairs fail to be formed in vivo. Such knowledge is important to understand the interplay between the formation of a functional BCR and the development of the antibody response, versus the possible autoimmunity arising from defects of this process[51,52]. Therefore, deciphering the molecular rules governing the pairing of H and L chains will benefit both basic B cell immunology and antibody discovery.

The issue of whether H–L antibody pairing specificity is predictable has been under debate for the past few decades. Seminal antibody structural analyses highlighted the interaction between the H and L chains at the antigen-binding site, consisting mainly of hypervariable regions in each chain known as the complementarity determining region (CDR) (Fig. 1a)[53,54]. Contact between the H and L chains is crucial to maintain antibody stability, as well as the orientation of the antigen-binding site[55–58]. The H–L interface is formed by contacts in the CDR region as well as the antibody framework region (FWR). Molecular biology experiments have identified FWR mutations, which would alter H–L interaction geometries and consequently abolish antigen binding[55,57]. Furthermore, mouse models coexpressing engineered H and L chains often further edit the L chains by introducing mutations which enhance the viability of B cells[59–61]. Computationally, earlier analyses focused on observations of nonrandom, over-represented H–L chain partners; however, statistical power to identify such associations were limited by the small number of paired H–L sequences and structures available for this type of analysis[62–64]. The development of single-cell sequencing methods, where single B cells are isolated, followed by extraction and sequencing of their H and L chain transcripts[36–38], has provided new insights to this problem. For instance, statistical analyses in DeKosky et al.[65] suggested that H–L pairing preference was random; however, others argued this could be due to idiosyncrasies in the experimental protocol[38]. More recently, a growing body of evidence suggested coherence between H and L chain choices in B cells. A study by Jaffe et al.[39] using newer single-cell methods found that H chains in mature, antigen-experienced B cells tended to use more restricted L chain partners than their naive counterparts. Furthermore, comparisons between artificial intelligence (AI) models trained on paired antibody H–L sequences posited that such models trained on paired data outperformed those trained on unpaired chains, in terms of learning biologically interpretable sequence embeddings and predicting antigen specificity[66]; moreover, given the H chain sequence, a stable L chain partner can be generated de novo[67]. Taken together, these findings suggest the existence of a set of molecular rules underlying the specificity of antibody H–L pairs. Tools for exploring these rules will hold the promise to understand and design better, more stable antibody pairs, facilitating the development of antibody therapeutics.

Here, we present ImmunoMatch, a suite of fine-tuned AI models for the classification of cognate antibody chain pairs. Based on an antibody-specific language model (AntiBERTa2[68]), ImmunoMatch was fine-tuned on full-length H and L chain variable domain sequences extracted from paired antibody repertoire data from healthy donors. This model outperformed baseline models using either CDR sequences or immunoglobulin gene usage as inputs. We found that further optimization to generate ImmunoMatch variants specific to antibody L chain types improved classification performance. We applied ImmunoMatch to study B cell development through the lens of optimizing H–L pairing, and identified chain pairing refinement as a hallmark of B cell maturation in both health and disease. We also validated ImmunoMatch in its ability to recover partner chains in therapeutic antibodies, and highlighted its ability to pinpoint important sequence patterns driving these predictions. Our results underscore the complexity in H–L chain pairing, and highlight the importance of chain pairing in understanding B cell development and engineering stable, functional antibodies.

## Results

### Machine-learning models for identifying cognate antibody chain pairing

We posited that machine-learning methods could allow us to test two competing hypotheses, namely that antibody H and L chain pairing preference can be predicted from sequence information, or that such pairing is random. Framing this as a binary classification task to distinguish between cognate, observed H–L pairs from randomly generated pairs, we curated paired H–L sequences, sampled from single-cell antibody repertoire datasets where their cell origin was barcoded as short nucleotide strings[40] (Fig. 1b). The coexistence of a H and a L chain with the same cell barcode was taken as evidence for paired chains, constituting our positive training examples. Owing to the removal of nonviable H–L pairs by natural selection[69], it was not possible to obtain negative training examples. We instead used a random shuffling strategy to generate 'pseudo-negative' examples, exchanging the light-chain partners between the observed, positive pairs. This procedure also guaranteed a balanced dataset with equal amounts of positive and pseudo-negative examples. Using three separate datasets[38,39,65] covering six donors, in total we curated 233,880 H–L pairs for training and testing, after balancing sample sizes over each donor to avoid bias due to the immunological background of individual donors.

We tested the contribution of different input features in combination with multiple machine-learning strategies. Analyzing antibody structural data[70,71], we observed that both the antibody FWR and the CDR3 contributed substantially to the interface between the variable heavy (VH) and variable light (VL) domains (Fig. 1c). Indeed, logistic regression and XGBoost models built solely on V and J gene usage achieved accuracies of 0.50 and 0.52, respectively, indicating limited predictive capability for heavy–light pairing preferences (Fig. 1d). To improve predictive performance, we next explored using CDR3 sequences as predictive features, in view of its substantial contribution to the VH–VL interface (Fig. 1c) and their high sequence diversity. We used a one-hot encoding approach and trained a convolutional neural network (CNN) with the CDR3 fragments of the H and L chains, leveraging its ability to capture local patterns within the data. Although the CNN model demonstrated moderate performance, attempts to further improve the model by changing the optimizer, incorporating additional convolutional layers or by adopting the ResNet[72] architecture, yielded minimal improvement (Fig. 1e). Taken together, these results highlighted the inherent limitations of only considering CDR3 or gene usage, potentially due to the lack of information on specific framework residues that participate in the VH–VL interface (Fig. 1c).

We therefore explored strategies to incorporate full-length VH and VL sequences in prediction, by capitalizing on recent advancement in protein language models[73], as well as those specifically trained using antibody sequences[68]. The transformer architecture, as employed in language models, excels at capturing long-range amino acid interactions[74,75]. Here, we compared an antibody-specific language model (AntiBERTa2; ref. [68]) against a generic protein language model (ESM-2 (ref. [73]), 150M parameters) (Fig. 1f). We observed that by fine-tuning ESM-2, its classification performance increased substantially, comparable to the performance of AntiBERTa2 before fine-tuning. The superior performance of AntiBERTa2 suggested that antibody-specific characteristics learned during pretraining were insightful for our task (Fig. 1f). Through these investigations of different machine-learning architectures (Fig. 1d–f), we used the fine-tuned model based on AntiBERTa2 as a final instance to classify antibody cognate H–L chain pairing. This model, ImmunoMatch, demonstrated an area under the receiver operator characteristic (AUC-ROC) of 0.75 (Fig. 1g). To further validate the generalizability of ImmunoMatch, we curated data from $n = 3$ donors unseen during

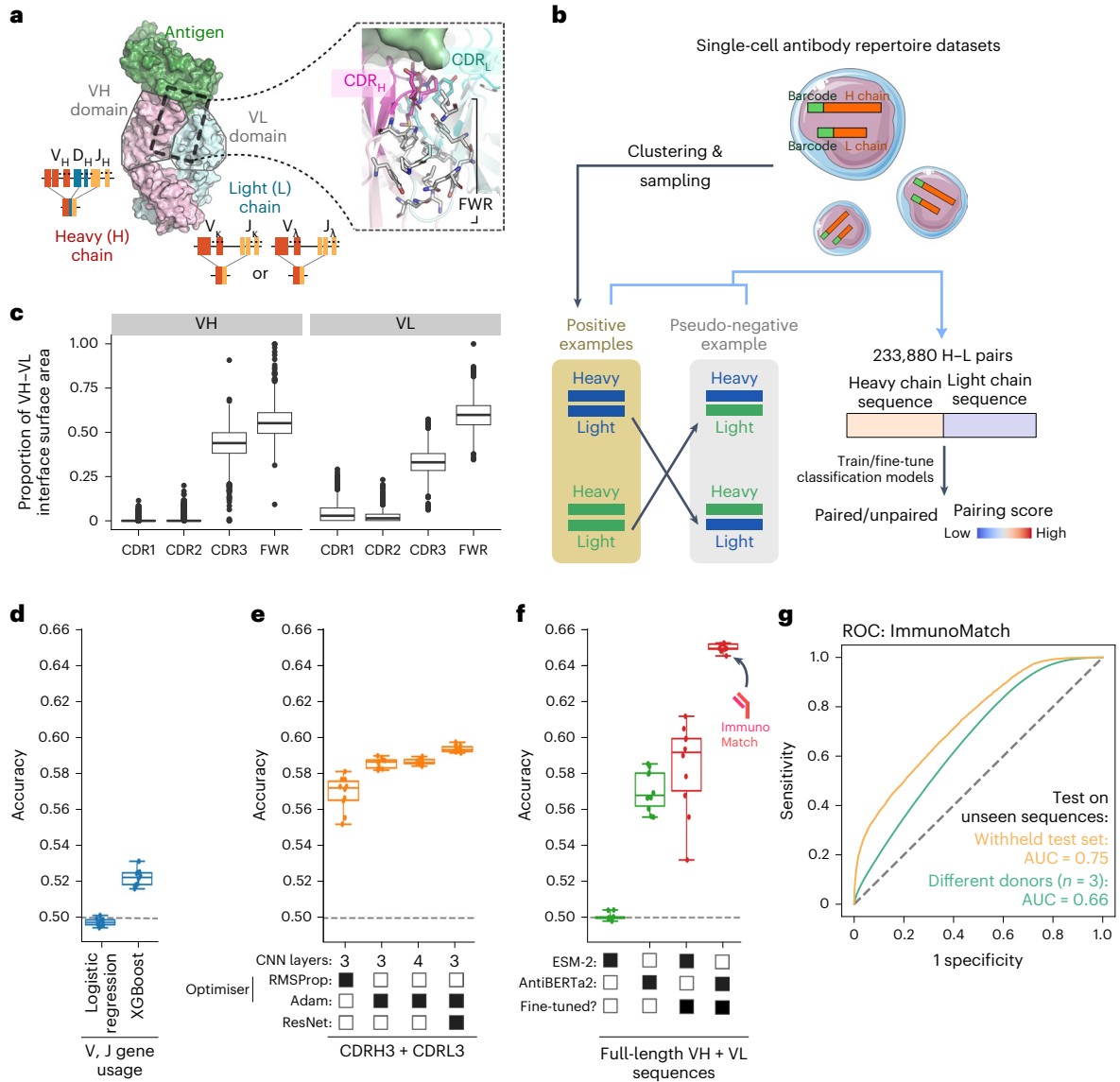

**Fig. 1 | ImmunoMatch for predicting cognate antibody chain pairing.**
**a**, Illustration of the antibody H–L interface (Protein Data Bank 6zlr), and its diversity potentiated by genetic recombinations in vivo. Inset on the right illustrates the H–L interface, with amino acid side chains (in sticks) to highlight positions in direct contact with the partner chains. CDR, complementarity determining region. **b**, Schematic of model training using curated positive and pseudo-negative examples from single-cell antibody repertoire datasets. See main text for further details. **c**, Proportion of the total surface area of the interface formed between the VH and the VL, contributed by individual CDR loops and the FWR, for $n$ = 3,781 human antibody structures[70].

The box-and-whisker plots depict distribution medians, lower and upper quartiles, and 1.5 × interquartile range. **d**–**f**, Accuracy of models trained solely on H and L chain V and J gene usage (**d**), one-hot-encoded CDRH3 and CDRL3 sequences (**e**) and full-length VH and VL sequences (**f**). Each data point represented a separate fold from the tenfold validation strategy employed during training (Methods), tested on the withheld test set ($n$ = 23,388). The box-and-whisker plots depict distribution medians, lower and upper quartiles, and 1.5 × interquartile range. The final ImmunoMatch model is indicated in **f**. **g**, ROC of the final ImmunoMatch model, calculated on the withheld test set (AUC = 0.75) and an external test set constituted by $n$ = 3 donors unseen during training (AUC = 0.66).

both pretraining and fine-tuning from Jaffe et al.[39]. Testing Immuno-Match on this external evaluation dataset, ImmunoMatch has an area under the receiver operating characteristic curve (AUC-ROC) of 0.66 (Fig. 1g). Details on other performance metrics of ImmunoMatch can be found in Table 1. We further confirmed that model performance was not impacted by immunoglobulin V and J gene usage (Extended Data Fig. 1a), suggesting that ImmunoMatch has learnt features beyond gene usage in H–L pairing prediction. We also validated that the combination of donors used here for training is informative for the model to learn pairing preferences (Supplementary Note 1, Supplementary Figs. 1–3 and Supplementary Table 2). Altogether, this suggests that ImmunoMatch can distinguish cognate antibody VH–VL pairing from randomly paired chains.

## ImmunoMatch performance could be further optimized via a light-chain-specific training strategy

We investigated whether the performance of ImmunoMatch can be further improved. Human antibodies use either one of the two types of light chains, $\kappa$ and $\lambda$, which are encoded by distinct DNA fragments located on separate chromosomes in the human genome[5]. The VL domains encoded by $\kappa$ and $\lambda$ genes are substantially different, as evidenced by pairwise sequence comparison of $n$ = 3,832 antibody structures[70] (Fig. 2a): on average $\kappa$ VL domains share 47.8% sequence identity with $\lambda$ VL domains, lower than the average identity of 69.1% within $\kappa$ light chains, and 61.8% within $\lambda$. We therefore hypothesized that training separate models on VH-V$\kappa$ and VH-V$\lambda$ sequence pairs could learn pairing patterns specific to either type of light chains.

**Table 1 | Performance metrics of ImmunoMatch and its variants**

|  | ImmunoMatch | ImmunoMatch-$\kappa$ | ImmunoMatch-$\lambda$ |
|---|---|---|---|
| F1 score | 0.677 | 0.839 | 0.796 |
| Accuracy | 0.666 | 0.817 | 0.764 |
| MCC | 0.333 | 0.660 | 0.556 |
| Precision | 0.655 | 0.747 | 0.700 |
| Recall | 0.701 | 0.958 | 0.922 |
| AUC-ROC | 0.753 | 0.885 | 0.831 |

MCC, Matthew's correlation coefficient.

Two specialized models, ImmunoMatch-$\kappa$ and ImmunoMatch-$\lambda$ (Fig. 2b), were fine-tuned using the same workflow as the original ImmunoMatch, with the sole exception that the models were only exposed to a light chain of one specific type during training.

We investigated the change in pairing scores from the original ImmunoMatch model and light-chain-specific models, for paired VH–V$\kappa$ and VH–V$\lambda$ accordingly (Fig. 2c,d). We noticed that there were some antibodies about which the original ImmunoMatch model was ambivalent (pairing scores peaked at around 0.5) for both H–$\kappa$ and H–$\lambda$ pairs. These peaks shifted toward pairing scores of 1 when ImmunoMatch-$\kappa$ (Fig. 2c) and ImmunoMatch-$\lambda$ (Fig. 2d) were used for prediction. Therefore, ImmunoMatch-$\kappa$ and ImmunoMatch-$\lambda$ are more specialized in capturing signals embedded in the sequences that are informative of H–L pairing. The performance of these variants of ImmunoMatch was further evaluated using separate datasets of antibodies with $\kappa$ and $\lambda$ light chains, which were withheld from fine-tuning. The distributions of pairing scores for their withheld test sets are shown in Extended Data Fig. 2, and a detailed summary of performance metrics of these models can be found in Table 1. ImmunoMatch-$\kappa$ achieved high accuracy (0.817) on $\kappa$ datasets, while ImmunoMatch-$\lambda$ performed comparably well on $\lambda$ datasets, with an accuracy of 0.764 (Fig. 2e and Extended Data Fig. 1b,c), both represent substantial improvements over the original ImmunoMatch (Table 1).

We also examined the generalizability of ImmunoMatch-$\kappa$ and ImmunoMatch-$\lambda$, by testing them on H–L pairs of light-chain types different to their respective training sets. When ImmunoMatch-$\kappa$ was tested on $\lambda$ datasets, we observed that this model could still achieve an accuracy above 0.5, albeit with performance decreased in comparison to the $\kappa$ test set (Fig. 2e). Of note, the performance of ImmunoMatch-$\lambda$ on $\kappa$ datasets remained largely unaffected by the differing distributions of the light-chain types between training and testing data (Fig. 2e). This suggests that ImmunoMatch-$\lambda$ is more generalizable in learning pairing rules for antibodies with $\kappa$ and $\lambda$ light chains. We further investigated the reason behind this by analyzing their confusion matrices, and found that ImmunoMatch-$\lambda$ made fewer false-negative predictions on H–$\kappa$ sequences, compared to applying ImmunoMatch-$\kappa$ on H–$\lambda$ pairs (Extended Data Fig. 3). This generalizability, caused by fewer false-negative predictions, may be linked to the process of B cell development in vivo (Fig. 2f). Initially, the heavy chains undergo gene rearrangement, followed by the formation of H–$\kappa$ pairs. They are then subject to central tolerance[69], which either removes B cells expressing unstable and autoreactive pairs of heavy and light chains by signaling them to cell death, or instructs them to rearrange the $\lambda$ gene locus to generate a H–$\lambda$ pair[9,76–78]. These H–$\lambda$ pairs are also subject to positive selection of B cells that express a stable H–L chain pair[46,79,80] and negative selection of those which react to self-antigens (Fig. 2f)[76]. If the H chain is able to pair with $\kappa$, the B cell will proceed in maturation, even though this H chain can theoretically form a pair with $\lambda$ as well. ImmunoMatch-$\kappa$ thus has difficulty in distinguishing between true negative and false negative H–$\lambda$ pairs; however, for an observed H–$\lambda$ pair, it implies that the H chain would have failed to pair with $\kappa$.

Therefore, ImmunoMatch-$\lambda$ is still able to capture the signals embedded in the sequence of negative H–$\kappa$ examples, leading to a low number of false-negative cases. In comparison to H–$\kappa$, H–$\lambda$ pairs represent a more homogeneous dataset encompassing H–L pairing features, leading to different performance of the two models (Discussion).

## ImmunoMatch facilitates pairing of heavy and light immunoglobulin chains in spatial transcriptomics data

The analysis above demonstrated that using ImmunoMatch-$\kappa$ and ImmunoMatch-$\lambda$ on H–$\kappa$ and H–$\lambda$ pairs respectively would be more accurate in H–L pairing prediction in comparison to the original ImmunoMatch model (Table 1). Using this approach, ImmunoMatch can be used to score and predict whether H and L chains given by the user form a cognate pair. This makes ImmunoMatch useful for comparing different single-cell BCR library preparation methods in their fidelity to generate well-resolved paired sequences (Supplementary Note 2, Supplementary Table 3 and Supplementary Fig. 4); however, there still remain data types that do not yet have single-cell resolution, and therefore necessitate H–L pairing annotation. A good example of this is spatial transcriptomics, where spatial VDJ methods have been described[81] (amplifying VDJ sequences from 10x Genomics Visium slides for long-read sequencing). Spatial methods are increasingly adopted in cancer and tissue immunology studies, but here the Visium protocol precludes direct identification of paired H–L chains, which will be necessary for downstream analysis such as antibody production to investigate their specificity.

In the original spatial VDJ manuscript, the authors proposed to predict H–L pairs by examining, in the spatial data, the colocalization of heavy and light-chain clones within the same tissue section[81]. Here we believe that ImmunoMatch can complement this method, going beyond comparing the transcript counts of the clones, to consider the complementarity of the full-length VH and VL sequences. We analyzed two breast tumor samples presented by Engblom et al.[81], which have been profiled using a multiregion Visium-based protocol and single-cell BCR sequencing. Applying ImmunoMatch-$\kappa$ and ImmunoMatch-$\lambda$ on their respective light-chain types, we first validated the performance of our models to correctly identify paired H–L chains in the single-cell libraries from these tumor samples (Fig. 3a). We then applied ImmunoMatch on the spatial VDJ data, and observed that ImmunoMatch pairing score successfully identified H–L pairs from the spatial data which overlapped with the single-cell data (Fig. 3b), especially when used in conjunction with the colocalization-based method ('repair') presented by Engblom et al.[81]. We further examined the H–L pair predictions on the tumor slides, and observed that ImmunoMatch can complement the Engblom et al. 'repair' method to predict H–L pairs from intratumoral B cells (Fig. 3c and Extended Data Fig. 4). Since ImmunoMatch directly considers the full-length VH and VL sequences, it addresses the pairing problem from an orthogonal perspective compared to identifying colocalized H and L chain transcripts. We believe ImmunoMatch therefore could be potentially used to facilitate the identification of cognate H–L pairs for antibody discovery applications in tissue immunology.

## Refinement of immunoglobulin chain pairing is a hallmark of B cell maturation

We next asked whether chain pairing likelihood would vary across stages of B cell development. The classical theory of B cell maturation posits that upon activation, naive B cells enter the germinal center (GC) to edit and optimize their BCRs to specifically bind their cognate antigens, with the successful binders subsequently exiting the GC and differentiating into memory B cells[82–84]. We collected paired H–L sequences from naive, GC and memory B cells from published studies[39,85], and scored the H–$\kappa$ and H–$\lambda$ sequences with ImmunoMatch-$\kappa$ and ImmunoMatch-$\lambda$ respectively. Comparing the pairing scores from these ImmunoMatch models between the B cell

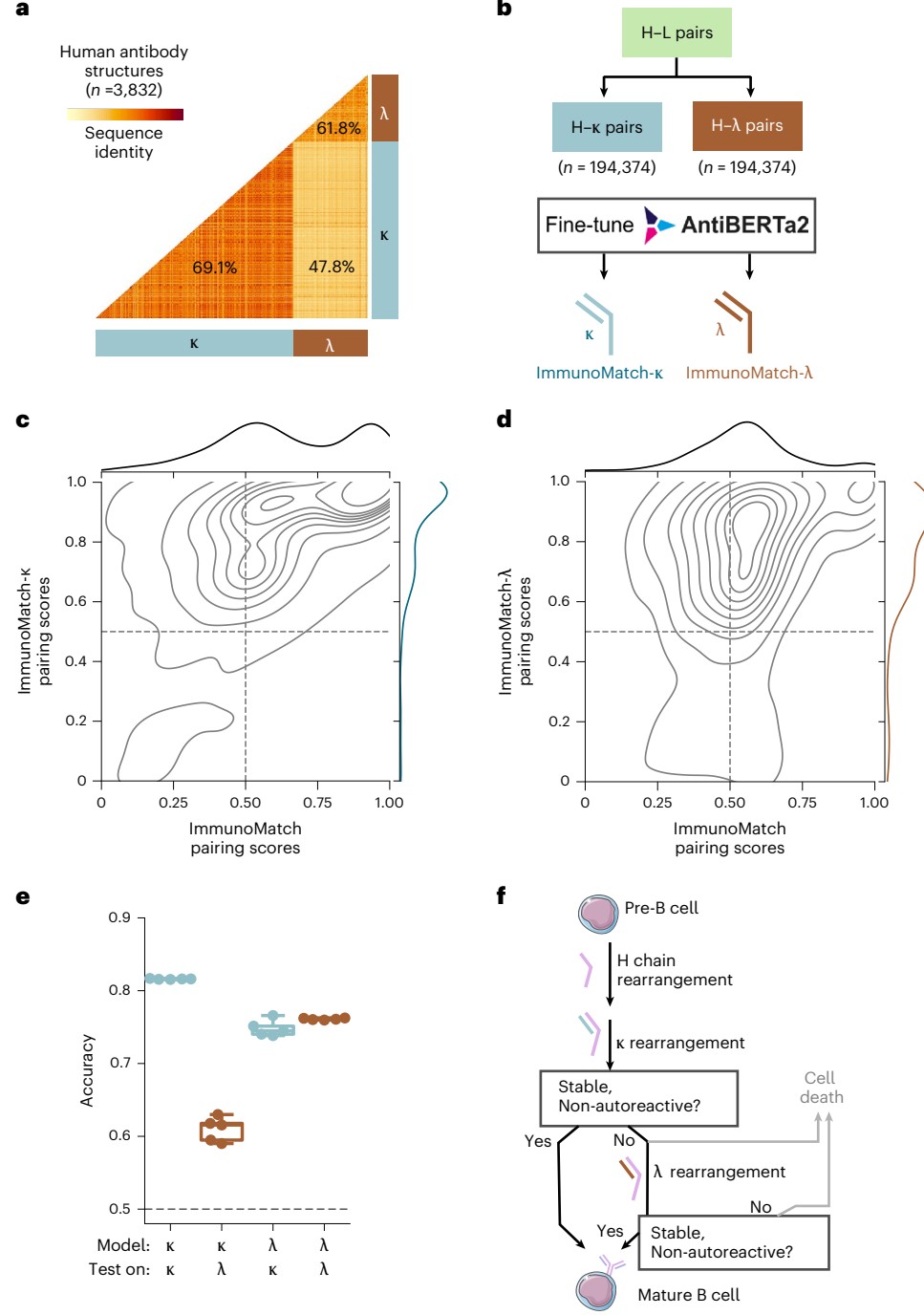

**Fig. 2 | An L chain-specific training strategy of ImmunoMatch was consistent with the in vivo mechanism of B cell development. a**, Sequence identity comparison between VL sequences taken from human antibody structures (*n* = 3,832) utilizing the *κ* and *λ* light chains. The averaged sequence identity within *κ* sequences, within *λ* sequences, and between *κ* and *λ*, are given on the plot. **b**, Strategy to extract H−*κ* and H−*λ* pairs from publicly available datasets to train separate ImmunoMatch-*κ* and ImmunoMatch-*λ* models. **c**, Pairing scores of H−*κ* pairs calculated by ImmunoMatch and ImmunoMatch-*κ*. **d**, Pairing scores

of H−*λ* pairs calculated by ImmunoMatch and ImmunoMatch-*λ*. **e**, Accuracy of ImmunoMatch-*κ* and ImmunoMatch-*λ* on withheld test sets comprised solely of H−*κ* and H−*λ* paired sequences. Each data point corresponded to a separate training fold in the cross-validation framework, tested on a withheld test set (*n* = 21,598). The box-and-whisker plots depict distribution medians, lower and upper quartiles and 1.5 × interquartile range. **f**, Schematic to illustrate the formation of the BCR in vivo.

subsets, we observed that memory B cells have substantially higher pairing score than their naive counterparts, with the distribution of GC B cells positioned between the two (Fig. 4a). We further defined clonally related naive and memory B cells based on CDRH3 sequence similarity, and identified examples of clonal expansions where pairing score increased as the clonotype diversified from the germline origin (Fig. 4b). We propose this continuum of chain pairing likelihood to be a

feature of B cell maturation: as BCRs undergo class-switch recombination and somatic hypermutation, H−L chain pairing is refined together with these processes, both integral in B cell maturation[17]. To test this hypothesis, we utilized a single-cell RNA sequencing (scRNA-seq) dataset of B cells sampled from the tonsil[85], and compared the H−L pairing score inferred using ImmunoMatch-*κ* and ImmunoMatch-*λ* for sequences of different heavy chain isotypes and mutational levels.

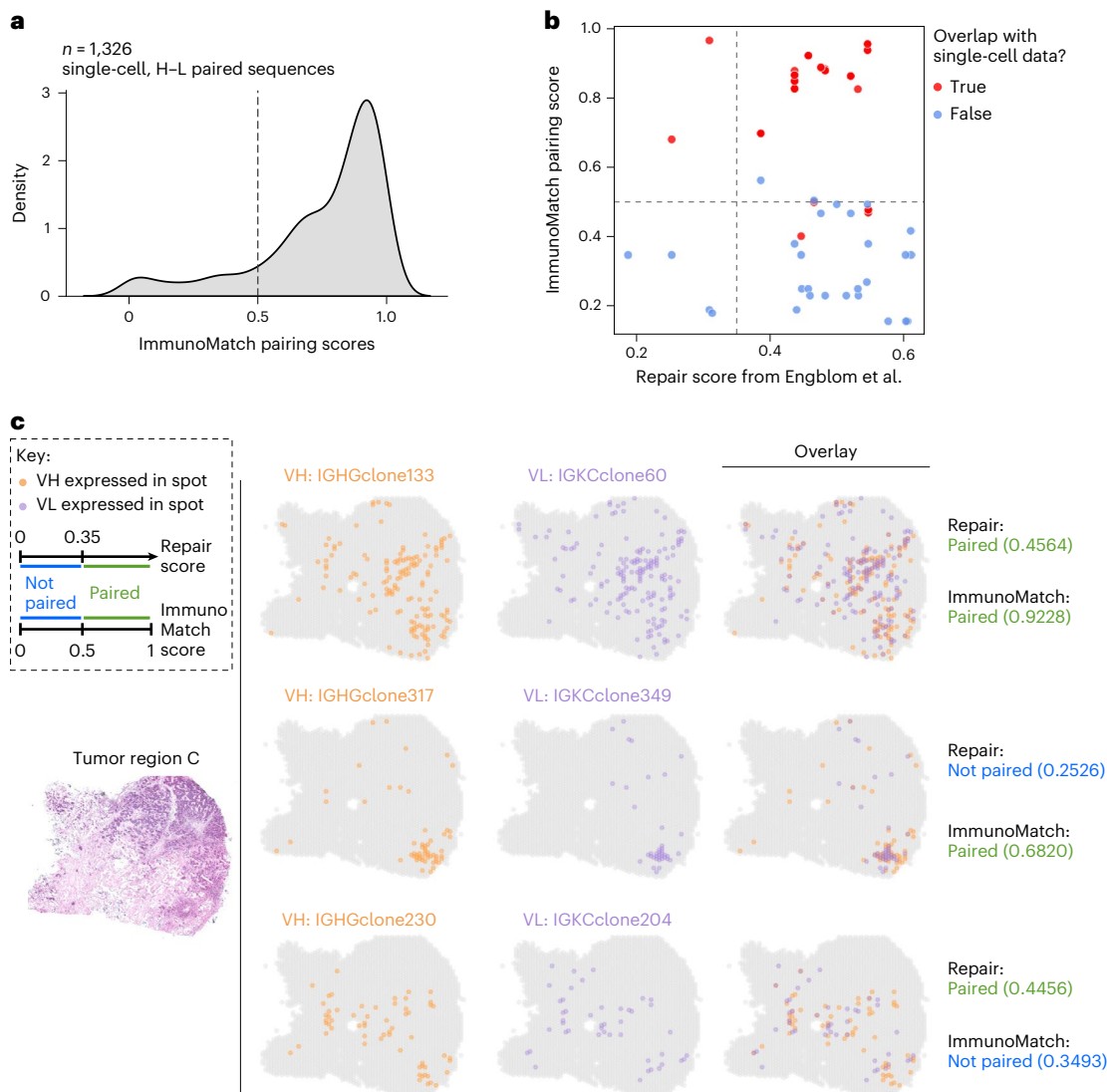

**Fig. 3 | ImmunoMatch facilitates pairing of heavy and light immunoglobulin chains in spatial transcriptomics data. a**, Distribution of ImmunoMatch pairing scores in single-cell, H–L paired sequences ($n = 1,326$) from $n = 2$ breast tumors analyzed in Engblom et al.[81]. **b**, Comparison between ImmunoMatch pairing score versus the 'repair' score from Engblom et al., for $n = 112$ H–L pairs reconstructed from spatial VDJ sequencing generated from 10x Visium profiling of breast tumor sections analyzed in Engblom et al. Each data point corresponds to one H–L pair, and grouped by whether the same CDRH3 and CDRL3 have been observed together with the same cell barcode in the single-cell library generated on the same sample. The decision boundary of ImmunoMatch and 'repair' method from Engblom et al. is indicated by the dashed line. **c**, Examples of H–L pairs analyzed by the 'repair' method of Engblom et al. and ImmunoMatch. The expression of VH and VL sequences in each H–L pair (row) across the analyzed tissue section is shown and overlaid on top of another. Each dot correspond to one spot in the 10x Visium slide. The tissue section hematoxylin and eosin (H&E)-stained image is shown for reference.

Of note, pairing score increases as the H chain switches away from IgM and IgD isotypes to IgG and IgA (Fig. 4c). Moreover, pairing scores display an inverse relationship with the H chain germline (Fig. 4c), independent of B cell subtypes (Extended Data Fig. 5). ImmunoMatch pairing scores therefore embed information about B cell maturation, and highlight the increase in H–L pairing specificity as a feature as BCR undergo maturation processes.

We further investigated whether a similar trend in the pairing scores can be found in BCRs isolated from diseases arising from B cell development. We collected $n = 123$ paired sequences from leukemia and lymphoma samples collated from the literature and publicly available databases of cancer cell lines, and mapped these samples to the different B cell developmental stages from which these cancers were thought to initiate[86,87]. Applying ImmunoMatch-$\kappa$ and ImmunoMatch-$\lambda$, we observed a continuum of H–L pairing scores for these sequences (Fig. 4d). Specifically, leukemia originating from pre-B cells in the bone marrow displayed a notably low pairing likelihood, reflecting their immature origin[88,89]. In contrast, in agreement with the need of a functional BCR for B cell activation and antigen interactions in these cancers[90,91], lymphoma samples typically displayed high pairing scores. These analyses suggest that ImmunoMatch models can be used to annotate immunoglobulin chain pairing, and that the refinement of chain pairing preference is a hallmark of B cell maturation in both health and disease.

## ImmunoMatch is sensitive to sequence differences in therapeutic antibodies

We finally investigated whether our ImmunoMatch models can be applied in an antibody discovery context. Specifically, we simulated an antibody triaging application, where ImmunoMatch-$\kappa$ and ImmunoMatch-$\lambda$ was used to score a random library of germline recombinations of H chain V and J gene segments against the cognate L chain partner (Fig. 5a). We performed this experiment on $n = 625$ therapeutic

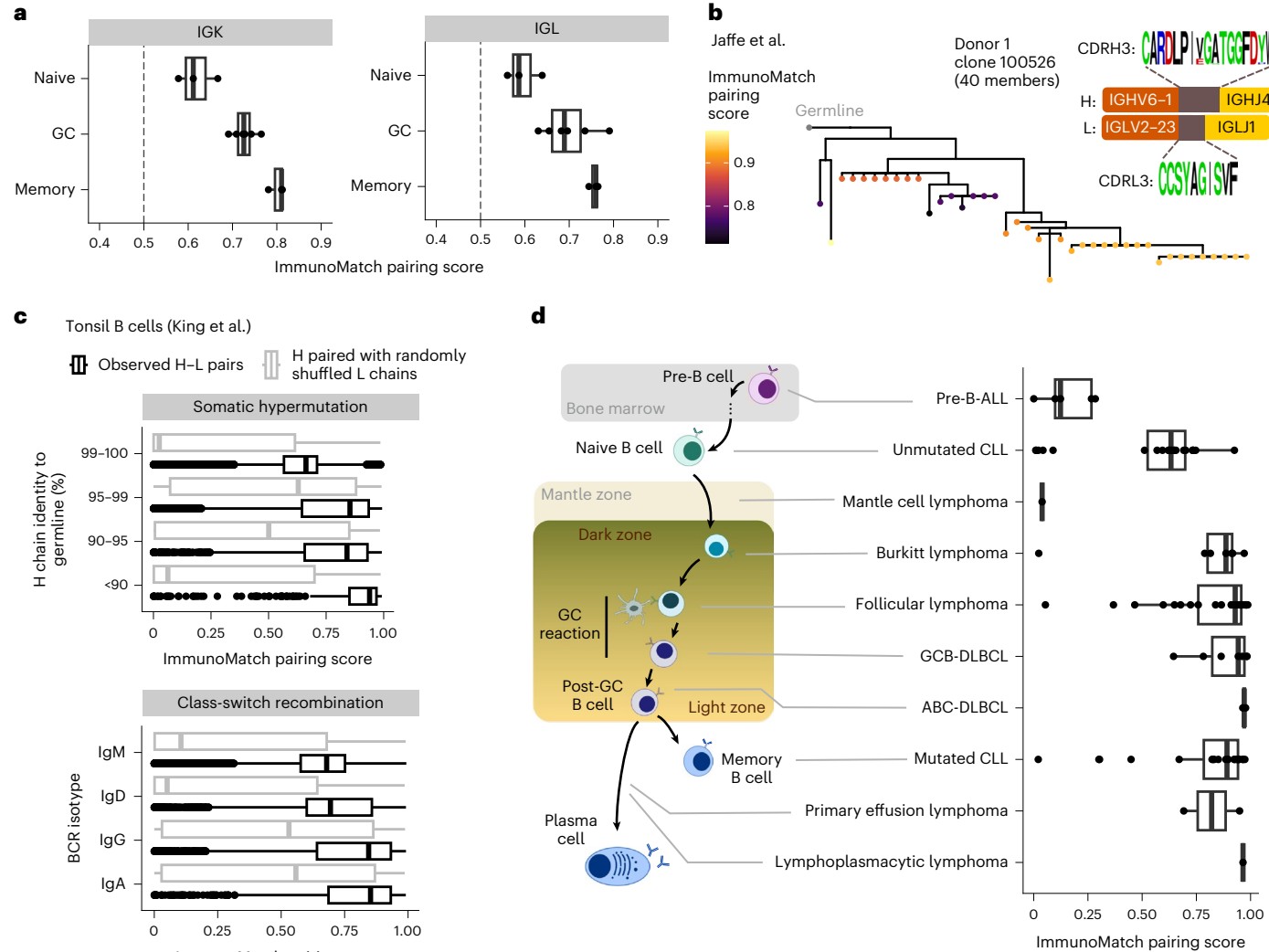

**Fig. 4 | ImmunoMatch revealed a continuum of BCR chain pairing likelihood across B cell development stages.** The box-and-whisker plots depict distribution medians, lower and upper quartiles and 1.5 × interquartile range. **a**, Pairing scores of BCRs from naive B cells ($n = 3$ donors), GC B cells ($n = 6$ donors) and memory B cells ($n = 3$ donors). Each data point represents average score per donor, calculated separately for H–$\kappa$ (panel 'IGK', scored using the ImmunoMatch-$\kappa$ model) and H–$\lambda$ ('IGL', scored using ImmunoMatch-$\lambda$) pairs. **b**, Example clonotype tree from Jaffe et al.[39] data with ImmunoMatch pairing scores mapped to individual observations as colored dots in the tree leaves. The germline configuration of VH and VL sequences are illustrated. **c**, ImmunoMatch pairing scores for paired VH and VL sequences from tonsil B cells ($n = 10,264$) in the King et al. dataset[85], grouped by their germline identity as a proxy of somatic

hypermutation status (top), or by the heavy chain isotype to illustrate class-switch recombination status (bottom). As a control, analogous annotation with ImmunoMatch was performed on for each cell, retaining the same observed VH sequence, but each paired with a randomly reshuffled L chain partner. **d**, Boxplots (right) depicting pairing scores of BCRs from leukemia and lymphoma samples ($n = 123$) curated from the literature (Methods). Data were organized by cancer subtypes and ordered by their corresponding B cell development stages when oncogenesis was thought to occur, according to published reviews[86,87] (schematic on the left). ALL, acute lymphoblastic leukemia; CLL, chronic lymphocytic leukemia; DLBCL, diffuse large B cell lymphoma; GCB, germinal center B cell; ABC, activated B cell.

antibodies, for which we generated random VH domain sequences, while preserving the observed CDRH3 fragment, for scoring against their cognate VL domains. We first verified that the ImmunoMatch pairing score was an effective discriminant ($P < 2 \times 10^{-16}$, Wilcoxon rank-sum test) of the observed H–L pairs in the therapeutic antibodies versus the random pairs (Fig. 5b). We hypothesized that the best random H–L pair from the ImmunoMatch models should resemble the wild-type sequence. Fig. 5c compares the best random VH match against the observed VH in terms of their sequence identities and ImmunoMatch pairing score difference. In line with our hypothesis, the higher the sequence identity, the more likely ImmunoMatch models were to output a similar pairing score. We however noted cases where the pairing score difference was substantial, despite sharing ≥80% sequence identity. We identified such cases and mapped the amino

acid positions where the randomly generated VH differed from the observed sequence (Fig. 5d). In these cases, the sequence differences typically reside in the CDRH1 and CDRH2 regions, and also at positions in the framework facing the VL domain (Fig. 5d). Given the importance of these positions in the VH–VL interface, this highlights the sensitivity of the ImmunoMatch models to structurally important positions to infer VH–VL chain pairing.

## Discussion

In this work, we identified an under-explored issue in antibody developability, namely antibody H–L chain pairing, and trained predictive models which specifically address this problem. We found that this was a tractable problem provided the right model architecture was sought, and the resulting modeling framework, ImmunoMatch,

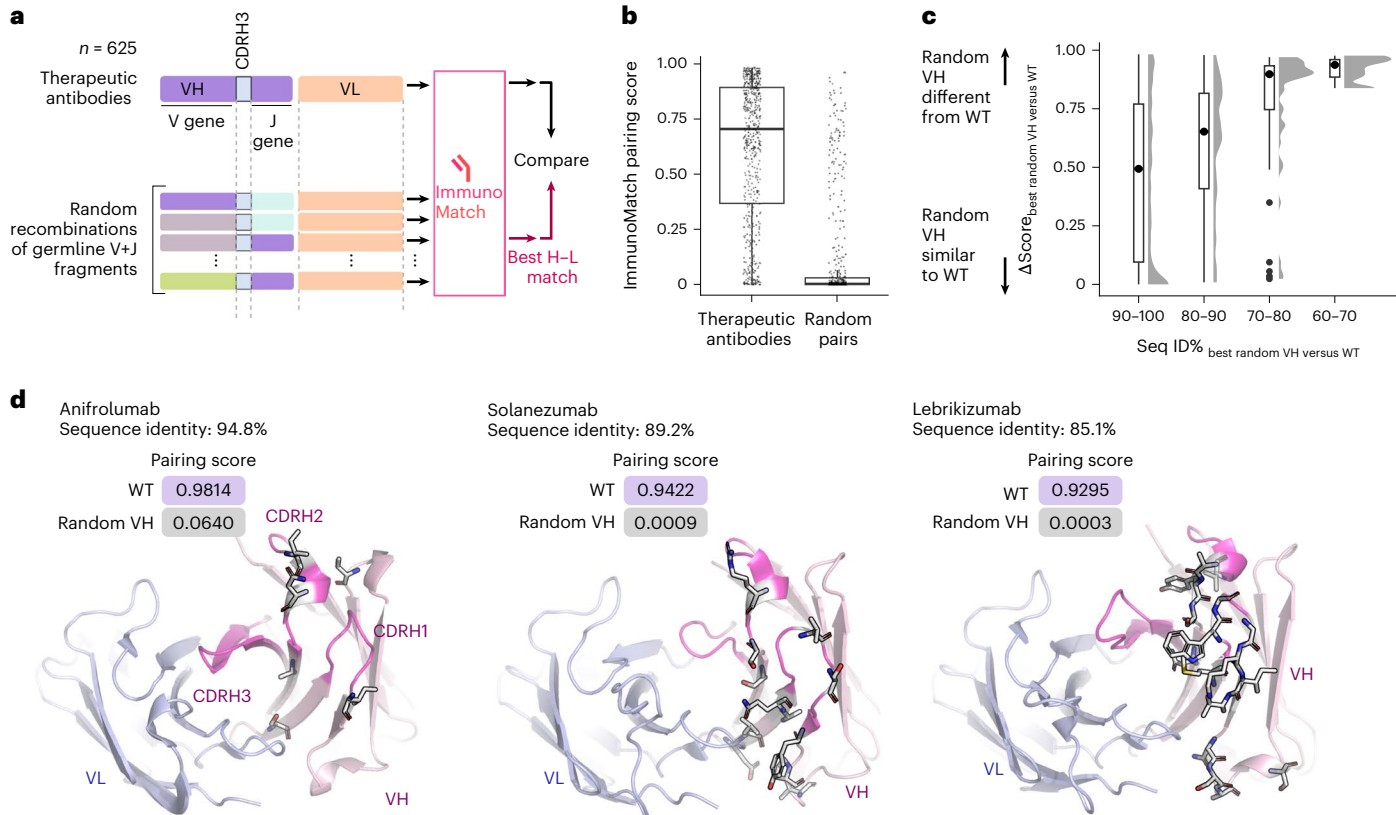

Sticks: residue differences between WT and VH sequence from best match

**Fig. 5 | ImmunoMatch is sensitive to VH–VL interface positions in therapeutic antibodies. a**, Outline of experiments performed on paired H–L chains from $n = 625$ therapeutic antibodies. For each antibody, we grafted the observed CDRH3 sequence onto random recombinations of H chain germline V and J fragments. Together with the observed L chain, these random H chains were subject to scoring by ImmunoMatch-$\kappa$ or ImmunoMatch-$\lambda$ (depending on L chain type). The random VH sequence from the best H–L match according to the model was compared against the observed VH sequence of the antibody. **b**, Comparison of ImmunoMatch pairing scores between therapeutic antibodies ($n = 625$) versus the pairs formed between the VL from these antibodies with randomly generated VHs ('random pairs'). The box-and-whisker plots depict distribution medians, lower and upper quartiles and 1.5 × interquartile range.

**c**, Absolute difference of the pairing scores between the observed (wild-type; WT) VH for therapeutic antibodies considered in this experiment ($n = 625$), and the randomly generated VH ranked best in H–L pairing using ImmunoMatch-$\kappa$ and ImmunoMatch-$\lambda$. A value of 0 means the pairing scores of the random VH and the WT are identical. The antibodies were binned (horizontal axis) by the sequence identity between the random VH and the WT. The box-and-whisker plots depict distribution medians, lower and upper quartiles and 1.5 × interquartile range. **d**, Examples of therapeutic antibodies where a large pairing score difference was observed between the random H–L pair to the true pair, despite high sequence identity between the VH sequences. Amino acids highlighted in sticks depict mismatched positions between the random VH and the observed VH used in the therapeutic antibodies.

demonstrated notable improvement over the baseline and revealed insights into B cell development. ImmunoMatch has several direct, practical uses: it can be applied to facilitate the pairing of H and L chains in spatial transcriptomics data. Additionally, it can be used to assess the fidelity of newly generated single-cell antibody repertoire datasets. Given the intensive development of new methods for capturing high-quality paired immunoglobulin sequences from single cells at an increasing throughput[40,81,92–94], a reliable method to assess the validity of the resulting datasets is crucial. Although the ImmunoMatch models were trained on healthy repertoires, we have successfully applied the method on sequences sampled from a variety of diseases, including both hematological cancers and B cells within solid tumors. The increased application of single-cell technologies to study these conditions will provide more data to further support the applicability of ImmunoMatch in these contexts. The ability of ImmunoMatch to identify cognate H–L pairs opens up possibilities to investigate how different positions on either chains contribute to pairing specificity and antibody stability, which could facilitate the design of new therapeutic antibodies. More broadly, the ImmunoMatch models can be part of a comprehensive assessment of antibody developability. A growing number of computational predictors have been developed to predict

antibody solubility, immunogenicity and other characteristics based on their sequences[25,26,31,95,96]; here, ImmunoMatch can complement the array of existing predictors to streamline computational antibody design. One could couple ImmunoMatch predictions with a sequence sampling method to generate new H–L pairs with optimized pairing properties. Going forward, a unified AI model with a training objective combining these antibody developability measures will be valuable in harmonizing the optimization process of candidate antibody designs. Methods that combine predictions from pools of expert models already exist and have been applied in the evolution of proteins[97]. As antibody development is essentially an engineering process aiming to reach the Pareto front where all desirable characteristics of the antibody are optimized, a combined training objective will facilitate reaching this front by minimizing costly iterative processes looking to balance between different developability issues[98,99]. This is also interesting from a basic B cell biology perspective, as the development of a viable antibody repertoire in vivo also follows a similar paradigm, balancing between antigen specificity, cross-reactivity, self tolerance and stably assembled BCRs capable of sustaining B cell viability[69]. AI approaches which predict these different aspects of BCR biology can be potentially combined to simulate B cell maturation, opening up new avenues in

investigating the fundamental pathological mechanisms in diseases where B cell maturation is disrupted.

An important distinction of ImmunoMatch is that its design principles are grounded on basic B cell biology. We reasoned that separate $\kappa$ and $\lambda$ models should produce superior performance compared to the original ImmunoMatch model, supported by the biological process to generate antibody L chains in the bone marrow. It is surprising that the ImmunoMatch-$\lambda$ model could accurately predict both H–$\kappa$ and H–$\lambda$ pairing; we have identified the reason for this was the reduced number of false negative predictions in the ImmunoMatch-$\lambda$ model when tested on H–$\kappa$ pairs. This mirrors the in vivo mechanism of B cell development, where the $\kappa$ locus is first rearranged to produce the light chain, followed by $\lambda$ if the H–$\kappa$ pair was eliminated during central tolerance. This secondary, 'rescue' role of $\lambda$ light chains during light-chain development[100,101] implies that many H–$\lambda$ pairs are results of failed $\kappa$ rearrangements, potentially facilitating ImmunoMatch-$\lambda$ but not ImmunoMatch-$\kappa$ to learn generalizable rules of chain pairing. Compared to the observed H–$\lambda$ pairs, H–$\kappa$ pairs represent a more heterogeneous data source, comprising both H chains which could only specifically pair with $\kappa$ light chains, and those which were ambivalent of the type of light-chain partner, but were coupled with a stable $\kappa$ sequence at the first attempt and did not have to undergo $\lambda$ rearrangement. On the other hand, H–$\lambda$ is less heterogeneous, potentially allowing fine-tuning to learn generalizable rules of H–L chain pairing. Additionally, we designed our randomization experiments on therapeutic antibodies simulating the assembly of the B cell receptor in vivo, screening for successful H–L pairs from libraries of randomly paired antibody chains, and observed a full spectrum of pairing likelihoods given the specifically chosen H and L chains. A number of computational methods aim to simulate antibody repertoires for benchmarking applications, or for learning the statistics of germline gene usage from repertoire data[102–104]. Our results suggest that simulation without constraining for paired H–L gene usage could substantially oversample dysfunctional sequences incapable of contributing to stable H–L pairs, which would call into question the reliability of these approaches in generating useful benchmarking single-cell, paired-chain datasets resembling real-life antibody repertoires. In general, most approaches to interrogate the functional antibody repertoire have been characterizing these molecules at the nucleic acid level, which is known to poorly correlate with protein-level measurements[105,106]. Going forward, mass spectrometry-based antibody repertoire sequencing would allow for direct assessment of observed antibody H–L protein chains[107–110] to sample well-expressed, stable H–L protein complexes, which is more relevant to our problem.

Predicting H–L chain pairing is highly complex, most notably evidenced by the performance ceiling of the original ImmunoMatch model trained on a mix of H–$\kappa$ and H–$\lambda$ sequences. We have explored possible causes of this limitation and found that this heterogeneity of light-chain type is part of the reason, as evidenced by the improvement in performance for the two ImmunoMatch variants specific to either light-chain type. However, there are several additional reasons which impose an implicit ceiling to the performance of ImmunoMatch: first, due to the limitation in sampling negative examples for training, we devised a strategy to generate pseudo-negative H–L pairs. These examples might actually pair in vivo, but the incomplete sampling of the repertoire prevented us from observing these pairs. This is reflected in the false positive instances observed after prediction, and our results demonstrate that ImmunoMatch (and the training data used to train the models) have mitigated this issue. Another possible reason is an in-built promiscuity in the number of possible partners of H and L chains. In our positive training examples we observed cases where one H chain could have as many as 1,000 distinct L chain partners. Similarly, earlier experimental investigations expressing different combinations of H and L chains often find promiscuous H chains which can be expressed with high yield together with a number of distinct L chains[34,35,111]. While

we know that antibody germline gene usage is highly biased toward a small handful of genes[112], we do not know whether the stability of H–L chain pairing contributes to skew this distribution. ImmunoMatch can be used to score specific versus promiscuous H–L pairs (identify the number of distinct H/L partners with which a given sequence can pair) to further investigate the sequence and molecular determinants of those examples which are ambivalent toward chain partners, and whether this promiscuity gives such chains a biological advantage to be more represented in the repertoire.

## Online content

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

## Methods

### Data curation

**Data acquisition.** Paired heavy and light-chain sequences were collected from six datasets across three publications, all derived from healthy human donors (Supplementary Table 1). Data from Rajan et al.[38] and DeKosky et al.[65] were downloaded following their associated publications, and the data from Jaffe et al.[39] were obtained from the Observed Antibody Space database[113]. The collected sequences represent diverse B cell populations, including naive, memory and mixed B cells. The use of multiple donors served to minimize potential biases in pairing preferences that could arise from donors' disease states or immunological background. These paired H−L sequences were annotated using IgBLAST[114] to identify germline gene usage and delineate the CDR3 regions for downstream analysis. These collected sequences were subject to the pipeline described below, involving clustering, sampling, and pseudo-negative generation, before training and testing machine-learning models.

**Clustering.** Clustering was employed to reduce over-representation of specific clonotypes in the training dataset and to prevent data leakage between the training and testing datasets. We extracted paired CDRH3 and CDRL3 (the most variable antibody segments defining clonotypes[115]), concatenated per pair and clustered them under a 90% sequence identity threshold, using the CD-HIT algorithm[116]. An equal number of heavy and light-chain pairs were sampled from each dataset to avoid biases stemming from individual data sources, with details summarized in Supplementary Table 1.

**Pseudo-negative data simulation.** Nonviable heavy and light-chain pairs are naturally eliminated during B cell development[69], implying a lack of negative labels to train machine-learning models. To address this, we simulated 'pseudo-negatives' by randomly selecting two H−L pairs and exchanging their light-chain partners. Given the variability in CDRL3 length, this randomization was further constrained by restricting that only H−L pairs with identical CDRL3 lengths were swapped. This procedure ensured that paired and pseudo-unpaired instances contributed equivalent CDR3 length features, and eliminating signals on H−L chain pairing originating from CDR3 length differences. Newly generated pseudo-negative pairs which were also present in the collected positive instances were rejected in the process, and another positive H−L pair would be sampled to exchange their L chain partners. A true H−L pair would be removed from the pool of positive examples if within 250 trials it failed to exchange the L chain with other pairs, to form a pseudo-negative example fulfilling the aforementioned criteria.

Following these steps, the dataset comprises 233,880 sequences with equal amount of positive and negative cases. The pre-processed data are split into training (90%) and testing (10%) subsets for model training and evaluation.

### Baseline models

In establishing all baseline models, these models were trained using a $K$-fold cross-validation approach ($K = 10$).

**Gene usage-based approach.** V and J gene usage was one-hot encoded and used as input for logistic regression and tree-based XGBoost models. Logistic regression was fitted using the LogisticRegressionCV function in scikit-learn (v.1.5.0). XGBoost models were fitted using the XGBoost python package (v.2.1.0), with the function XGBClassifier and the parameter 'tree_method = hist'.

**CDR3-based approach.** We leveraged the CNN architecture to capture local patterns in CDRH3 and CDRL3 sequences. Briefly, CDRH3 and CDRL3 amino acid sequences were separately one-hot encoded as a two-dimensional $n \times L$ matrix, where $n$ is the length of the CDR3 fragment and $L$ represents the 20 amino acids. The two matrices were passed separately through convolutional layers followed by concatenating outputs from the two streams for passing through to a multilayer perceptron. Various hyperparameters were tested in training the CNN, including the number of convolutional layers, choice of optimizers and the incorporation of a residual network (ResNet)[72]. Further details are in Supplementary Methods.

**Language model-based approach.** Entire VH and VL paired sequences from the curated training set were passed into pretrained language models (ESM-2 (150M parameters)[73] and AntiBERTa2 (ref. 68)) appended with a classification head for binary classification (paired (positive) versus unpaired (pseudo-negative)). Pretrained weights of these models were obtained via HuggingFace. We compared the classification results between the fine-tuned models (for $n = 3$ epochs with learning rate $2 \times 10^{-5}$ and weight decay 0.01, using the Trainer method in HuggingFace) versus the pretrained models (that is weights were frozen during fine-tuning). Supplementary Methods provide further details.

### The final ImmunoMatch model

We compared the aforementioned model setups to identify the model with the highest accuracy in the withheld test set. The final model was selected as the fine-tuned AntiBERTa2 model. We call this fine-tuned instance ImmunoMatch. We noted that the withheld test sequences were sampled from the same donors exposed to ImmunoMatch during fine-tuning. We thus curated an external test set, comprising sequences from previously unseen donors in both pretraining and fine-tuning stages. We used paired H−L chains from donor 1, 2 and 4 from Jaffe et al.[39] as this external test set (donor 3 was included in training; Supplementary Table 1). Data were downloaded from the associated figshare repository (https://doi.org/10.25452/figshare.plus.20338177) of the Jaffe et al. publication. Annotated VH and VL sequences were obtained from the 'filtered_contig_annotations.csv' files processed from respective sequencing libraries.

### ImmunoMatch-κ and ImmunoMatch-λ models

We further produced two variants of ImmunoMatch, ImmunoMatch-$\kappa$ and ImmunoMatch-$\lambda$, by curating antibody heavy chain sequences with $\kappa$ and $\lambda$ light chains, respectively for fine-tuning AntiBERTa2. The previously collected paired antibody sequences were split into two groups according to their light-chain types, and separately subjected to clustering and pseudo-negatives generation steps. The sizes for training sets of ImmunoMatch-$\kappa$ and ImmunoMatch-$\lambda$ were kept constant ($n = 194{,}374$). ImmunoMatch-$\kappa$ and ImmunoMatch-$\lambda$ were trained using the same procedure and parameters as the original ImmunoMatch model.

### Applying ImmunoMatch models

For ImmunoMatch, ImmunoMatch-$\kappa$ and ImmunoMatch-$\lambda$, the epoch with the minimized evaluation loss was loaded for application on external datasets using the HuggingFace 'RoFormerForSequenceClassification.from_pretrained' interface. A pairing score was derived by applying the softmax transformation on the output obtained by passing through an H−L sequence pair. To apply ImmunoMatch in biological case studies, the annotated sequences were first grouped by their light-chain types, and the ImmunoMatch-$\kappa$ and ImmunoMatch-$\lambda$ models were applied on the H−$\kappa$ and H−$\lambda$ subsets, respectively.

### Validation datasets

**Spatial VDJ sequencing dataset.** We analyzed data generated from $n = 2$ breast tumors reported in Engblom et al.[81], where the VDJ sequences of intratumoral B cells were analyzed by (1) the authors' spatial VDJ sequencing protocol, and (2) single-cell VDJ sequencing data generated from the same tumor sections in parallel to (1). Both spatial and single-cell datasets were obtained from the Zenodo repository

associated to the Engblom et al. manuscript (https://doi.org/10.5281/zenodo.7961605). For the spatial data, the complete VDJ amino acid sequences for the heavy and light chains were extracted from the .vdjca files provided by the authors, using the exportAlignments function in MiXCR (v.3.0.3). The complete sequences were merged with the authors' clone annotations and results from the Engblom et al. 'repair' method described in their manuscript[81] based on identical CDRH3 and CDRL3 amino acid sequences. In total for the spatial data, we considered $n = 112$ H−L pairs with complete, functional amino acid sequences for both chains. The spatial locations of H−L pairs were visualized by using the 10x Genomics Space Ranger output and Seurat objects provided by the authors. For the single-cell data, we obtained the Cell Ranger output provided by the authors in the Zenodo repository, and analyzed a total of $n = 1,326$ pairs of complete, functional H−L sequences. Overlap between the spatial and single-cell data was determined by identifying identical CDRH3 and CDRL3 amino acid sequences.

**Healthy B cell single-cell datasets generated using different library preparation protocols.** We collected single-cell BCR pairs from healthy individuals sampled using three different single-cell library preparation methods: (1) SMART-seq2 data from Lindeman et al.[117]; (2) 10x Genomics example data provided by the manufacturer (https://www.10xgenomics.com/datasets/human-b-cells-from-a-healthy-donor-1-k-cells-2-standard-6-0-0); and (3) Parse Bioscience example data from the manufacturer (https://www.parsebiosciences.com/datasets/bcr-sequencing-of-1-million-healthy-and-diseased-samples-in-a-single-experiment/). The Parse Bioscience dataset contain B cells from both $n = 12$ healthy donors and $n = 12$ patients with autoimmune diseases; here only the data corresponding to the healthy controls were considered. For each dataset, nonfunctional and incomplete sequences were removed. The H and L sequences were grouped by unique cell identifiers and all possible combinations of H−L pairs with the same cell identifier were considered by Immuno-Match scoring. Supplementary Note 2 provides further details.

**Naive, germinal center and memory B cell datasets.** We collected the following repertoire datasets which comprised paired heavy and light-chain sequences traceable to the cell of origin. First, for naive and memory B cells from healthy individuals, we used data from donors 1, 2 and 4 in Jaffe et al.[39]. Here, only sequencing libraries containing purely sorted naive or memory (unswitched or switched) were considered. In total we considered $n = 711,372$ paired sequences from the three donors in the study. Second, we curated data from a single-cell study of tonsil B cells[85]. Quality filtered, annotated sequences were downloaded from EMBL-EBI ArrayExpress (accession code E-MTAB-9003) and overlapped with cell-type annotation available for the same dataset based on scRNA-seq[85] (ArrayExpress accession code E-MTAB-9005). We used the King et al. dataset for two analyses: first, we extracted sequences corresponding to GC B cells by considering the associated cell cluster labels ('GC', 'LZ GC', 'DZ GC' and 'FCRL2/3high GC'), for comparisons with the naive and memory B cell data from Jaffe et al. This amounted to $n = 1,823$ paired sequences. Second, we used the entire King et al. dataset to examine the differences in ImmunoMatch pairing scores across different heavy chain isotypes and germline identity levels. For this we considered $n = 10,782$ cells with paired VH and VL sequences, and generated pseudo-negative pairs as control using the identical procedure described in the section 'Pseudo-negative data simulation'. Clonotype clustering analysis was performed on the Jaffe et al. dataset using the DefineClones.py function in the Change-O (v.1.3.0)[118] package. Clonotype trees were constructed using the maximum parsimony method 'dnapars' in PHYLIP[119], accessed through BrepPhylo[120].

**B cell leukemia and lymphoma datasets.** We compiled paired heavy and light chains from B cell leukemia and lymphoma samples, from two

sources: (1) searches on the GenBank database, which retrieved $n = 92$ paired heavy and light chains obtained from leukemia and lymphoma samples reported in published works[121–127], and (2) leukemia/lymphoma cell lines from the Cancer Cell Line Encyclopaedia (CCLE) project[128]. For (1) sequences were annotated using IgBLAST[114] (v.1.19.0) with germline sequences obtained from IMGT (accessed on 22 November 2024). For (2), paired-end bulk RNA sequencing FASTQ files were obtained from the Sequence Read Archive (project accession PRJNA523380), and heavy and light-chain sequences were assembled following Tan et al.[129], using MiXCR (v.4.7.0) software[130] run with default parameters. The following filtering steps were performed to ensure that paired sequences were from cancers of a B cell origin: (1) removed cell lines with at least 100 reads mapped to T cell receptors; (2) removed cell lines with fewer than 100 reads mapped to immunoglobulins; and (3) removed cell lines without any heavy chain reads. We noted in these cases, more than one distinct heavy and light-chain sequences can often be found[129]. We therefore implemented the following procedure to obtain a single H−L pair for each cancer cell line. First, we examined the ratio between the total number of reads mapped to the IGK locus to that for the IGL locus; we only considered the specific light-chain type if there were at least 100 times more reads mapped to this light-chain locus. Second, only the heavy and light chains with the highest fraction of read support were retained. Following removal of out-of-frame sequences we analyzed $n = 31$ cancer cell lines from the CCLE project. In total we analyzed $n = 123$ paired heavy and light chains from B cell leukemia and lymphoma.

## Antibody structure analysis

We analyzed $n = 3,832$ human antibody structures curated in VCAb[70]. VH−VL interface surface area was calculated by estimating the change in solvent-exposed surface area upon VH−VL complex formation, using POPSCOMP[71]. Definitions of VH and VL domains were taken from VCAb annotations. CDR and FWR regions were delineated using IMGT numbering obtained using ANARCI[131]. For comparisons between $\kappa$ and $\lambda$ light chains, sequence identities were computed using MMseqs2 (v.13.45111)[132].

## Analysis of therapeutic antibodies

We downloaded therapeutic antibody annotations from TheraSAb-Dab[133] (accessed 5 December 2024), and considered only those which were (1) monospecific (with only one unique H−L pair); (2) approved or under active development; and (3) either human or humanized. Sequences were numbered using ANARCI[131]. For each antibody, we kept the observed VL sequence as it was; for VH, we grafted the observed CDRH3 fragment onto all possible combinations of germline *IGHV*- and *IGHJ*-encoded amino acid sequences obtained from IMGT (5 December 2024). This constituted a library of random VH sequences while retaining the same CDRH3. This library was screened together with the observed VL using ImmunoMatch to obtain pairing scores, for comparison with the pairing score of the observed VH−VL pair. Sequence identities were computed using MMseqs2 (ref. 132; v.13.45111).

## Reporting summary

Further information on research design is available in the Nature Portfolio Reporting Summary linked to this article.

## Data availability

Publicly available antibody repertoire sequencing datasets were utilized for training and testing ImmunoMatch: Rajan et al.[38] (SRP104286), DeKosky et al.[65] (PRJNA315079, SRX709625 and SRX709626), King et al.[85] (E-MTAB-9003), Lindeman et al.[117] (Supplementary Table 5 in the paper), Jaffe et al.[39] (authors' data repository at https://doi.org/10.25452/figshare.plus.20338177) and Engblom et al.[81] (authors' data repository at https://doi.org/10.5281/zenodo.7961605). Curated sequences from leukemia and lymphoma samples, as well as therapeutic antibody

sequences considered in this analysis, can be found in the project GitHub repository (https://github.com/Fraternalilab/ImmunoMatch). Source data are provided with this paper.

## Code availability

Final checkpoints of ImmunoMatch, ImmunoMatch-$\kappa$ and ImmunoMatch-$\lambda$ are available on HuggingFace at https://hugging-face.co/fraternalilab/immunomatch. Code to run ImmunoMatch to annotate sequences can be found on Google Collaboratory (https://colab.research.google.com/github/Fraternalilab/ImmunoMatch/blob/main/Run_ImmunoMatch.ipynb). A standalone Python package to apply ImmunoMatch is available on PyPI at https://pypi.org/project/ImmunoMatch/. Data and code to generate figures in this manuscript are available on GitHub (https://github.com/Fraternalilab/ImmunoMatch).

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

## Acknowledgements

We thank all members of the Fraternali group for comments and suggestions. This work was supported by the Biotechnology and Biological Sciences Research Council (https://bbsrc.ukri.org/, BB/T002212/1 to F.F., D.K.D.-W. and J.C.F.N.; BB/B000745/1 to F.F. and J.C.F.N.). D.G. was supported by a PhD scholarship from the China Scholarship Council (no. 202008440414). The funders had no role in study design, data collection and analysis, decision to publish or preparation of the article.

## Author contributions

D.G. curated training, testing and validation datasets, implemented ImmunoMatch and performed model comparisons, supervised by F.F., J.C.F.N. and D.K.D-W. J.C.F.N. and D.G. curated ImmunoMatch use cases and carried out computational analyses. F.F. conceived the project and acquired funding. J.C.F.N. and D.G. wrote the manuscript and the Methods section with critical input from F.F. All authors read, commented and approved the final manuscript.

## Competing interests

The authors declare no competing interests.

## Additional information

**Extended data** is available for this paper at https://doi.org/10.1038/s41592-025-02913-x.

**Correspondence and requests for materials** should be addressed to Franca Fraternali or Joseph C. F. Ng.

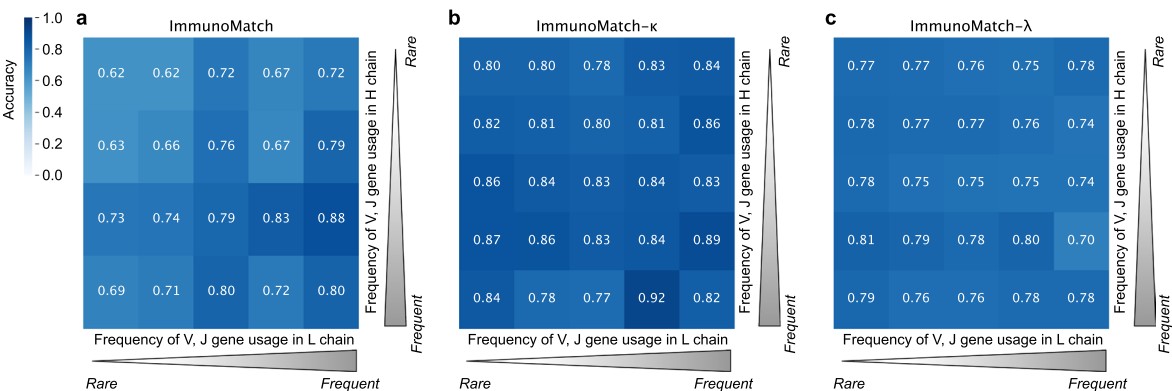

**Extended Data Fig. 1 | Prediction accuracy on withheld sequences, grouped by the frequency of V and J gene usage in the H and L chains.** For each chain we grouped sequences by their gene usage into five bins. Results were shown for (**a**) ImmunoMatch (n = 23,388), (**b**) ImmunoMatch-$\kappa$ (n = 21,598) and (**c**) ImmunoMatch-$\lambda$ (n = 21,598) models separately.

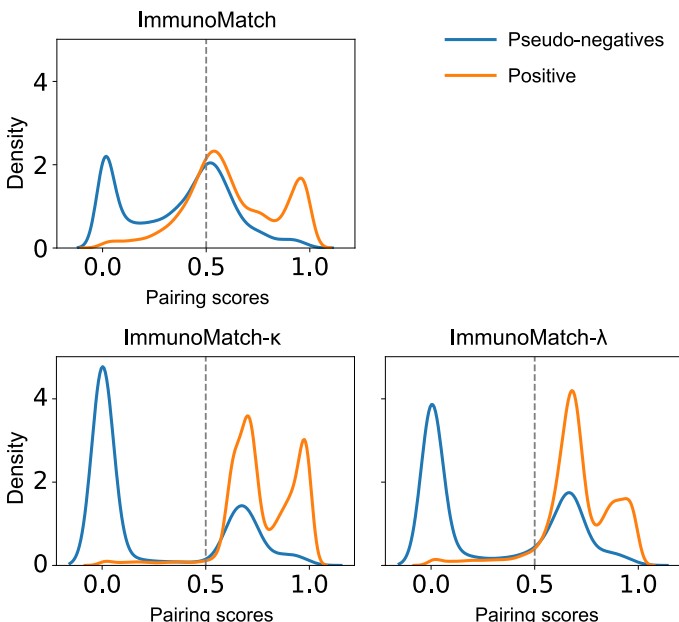

**Extended Data Fig. 2 | Distribution of pairing scores from withheld test sequences for the ImmunoMatch models.** Shown here data for the ImmunoMatch (top, n = 23,388), ImmunoMatch-$\kappa$ (bottom left, n = 21,598) and ImmunoMatch-$\lambda$ (bottom right, n = 21,598) models. The positive (orange) and pseudo-negative (blue) pairs were visualized with separate density curves.

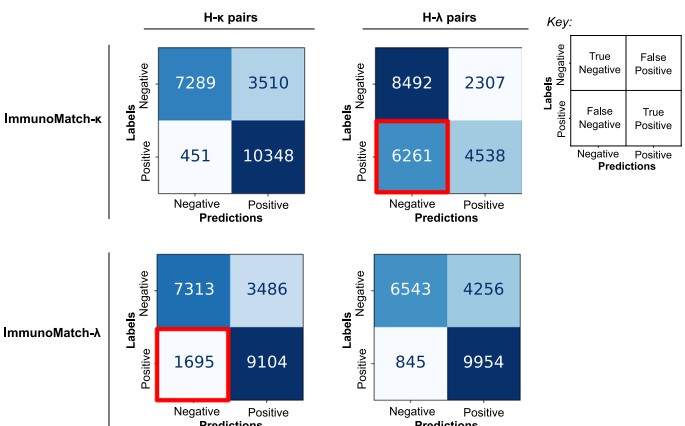

**Extended Data Fig. 3 | Confusion matrices of ImmunoMatch-$\kappa$ and ImmunoMatch-$\lambda$ on H-$\kappa$ and H-$\lambda$ sequences.** Data for ImmunoMatch-$\kappa$ and ImmunoMatch-$\lambda$ models were organized by rows, tested on H-$\kappa$ and H-$\lambda$ pairs (columns). False negative (FN) predictions when models are tested on datasets of different L types are highlighted.

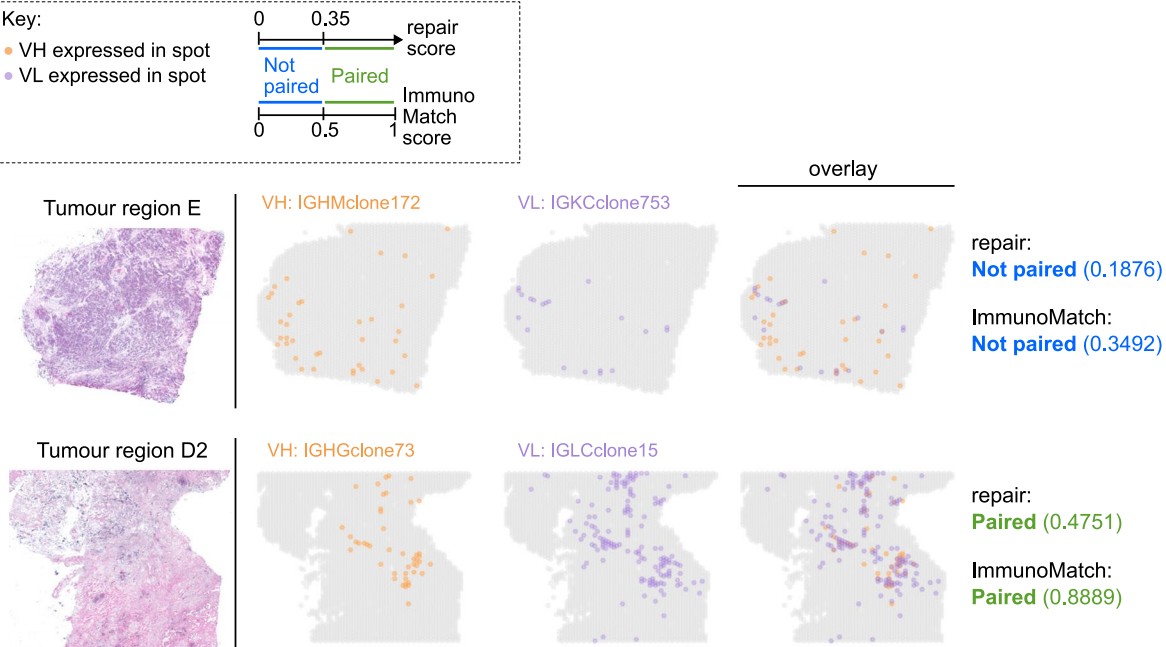

**Extended Data Fig. 4 | Additional examples of H–L pairs analyzed by Engblom et al.'s 'repair' method and ImmunoMatch.** The expression of VH and VL sequences in each H–L pair (row) across the analyzed tissue sections are shown and overlaid on top of another. Each dot corresponds to one spot in the 10X Visium slide. The tissue section hematoxylin and eosin (H&E) stained images are shown for reference.

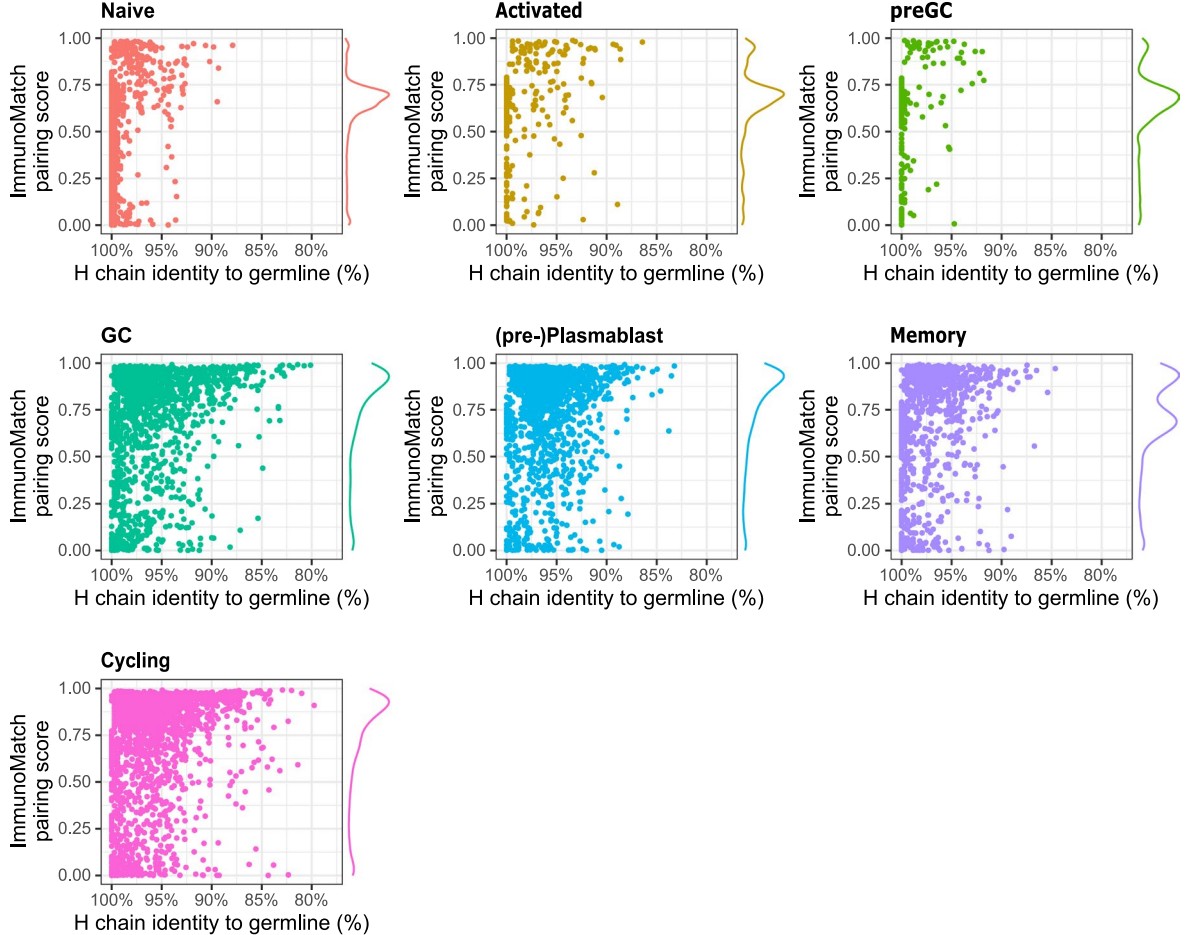

**Extended Data Fig. 5 | Relationship between somatic hypermutation level and ImmunoMatch pairing score in the King et al. tonsil dataset.** In each panel somatic hypermutation level (horizontal axis, represented as sequence identity to germline in the heavy chain) was plotted against ImmunoMatch pairing score (vertical axis). Data from the King et al. Tonsil single-cell BCR repertoire dataset (n = 10,264 cells) were shown. This relationship is statistically tested (p < 2e-16), using a mixed effect model to control for cell type (as a random effect).

Each data point corresponds to a single B cell (that is one paired H−L chain). Data for different cell types were visualized in separate panels. To simplify the visualization we grouped together multiple cell type labels corresponding to germinal center (GC) B cells (original labels in King et al.: 'LZ GC', 'GC', 'DZ GC' and 'FCRL2/3high GC'), (pre-)plasmablasts (original labels 'prePB' and 'Plasmablast') and memory B cells (original labels 'MBC' and 'MBC FCRL4+').

# Reporting Summary

## Statistics

For all statistical analyses, confirm that the following items are present in the figure legend, table legend, main text, or Methods section.

| n/a | Confirmed | |
|---|---|---|
| ☐ | ☒ | The exact sample size (*n*) for each experimental group/condition, given as a discrete number and unit of measurement |
| ☐ | ☒ | A statement on whether measurements were taken from distinct samples or whether the same sample was measured repeatedly |
| ☐ | ☒ | The statistical test(s) used AND whether they are one- or two-sided *Only common tests should be described solely by name; describe more complex techniques in the Methods section.* |
| ☐ | ☒ | A description of all covariates tested |
| ☐ | ☒ | A description of any assumptions or corrections, such as tests of normality and adjustment for multiple comparisons |
| ☐ | ☒ | A full description of the statistical parameters including central tendency (e.g. means) or other basic estimates (e.g. regression coefficient) AND variation (e.g. standard deviation) or associated estimates of uncertainty (e.g. confidence intervals) |
| ☐ | ☒ | For null hypothesis testing, the test statistic (e.g. $F$, $t$, $r$) with confidence intervals, effect sizes, degrees of freedom and *P* value noted *Give P values as exact values whenever suitable.* |
| ☒ | ☐ | For Bayesian analysis, information on the choice of priors and Markov chain Monte Carlo settings |
| ☒ | ☐ | For hierarchical and complex designs, identification of the appropriate level for tests and full reporting of outcomes |
| ☒ | ☐ | Estimates of effect sizes (e.g. Cohen's *d*, Pearson's *r*), indicating how they were calculated |

*Our web collection on statistics for biologists contains articles on many of the points above.*

## Software and code

Policy information about availability of computer code

| Data collection | The manuscript utilised the following paired antibody repertoire sequencing datasets: Rajan et al. (Commun Biol 2018), DeKosky et al. (PNAS 2016), Jaffe et al. (Nature 2022), Phad et al. (Nat Immunol 2022), James et al. (Nat Immunol 2020), Eccles et al. (Cell Rep 2020), King et al. (Sci Immunol 2021), Engblom et al. (Science 2023), Lindeman et al. (Nature Methods 2018), datasets from 10x Genomics (https://www.10xgenomics.com/datasets/human-b-cells-from-a-healthy-donor-1-k-cells-2-standard-6-0-0), Parse Biosciences (https://www.parsebiosciences.com/datasets/bcr-sequencing-of-1-million-healthy-and-diseased-samples-in-a-single-experiment/) and a curated dataset from individual lymphoma and leukaemia samples from the GenBank database. The data were collected and curated manually. |
|---|---|
| Data analysis | All code to use and apply the ImmunoMatch models, as well as scripts to generate the figures and tables in the manuscript , can be found in the GitHub repository (https://github.com/Fraternalilab/ImmunoMatch). ImmunoMatch is available as a standalone python package (https://pypi.org/project/ImmunoMatch/), as well as interactive notebook on Google Collaboratory (https://colab.research.google.com/github/Fraternalilab/ImmunoMatch/blob/main/Run_ImmunoMatch.ipynb). |

For manuscripts utilizing custom algorithms or software that are central to the research but not yet described in published literature, software must be made available to editors and reviewers. We strongly encourage code deposition in a community repository (e.g. GitHub). See the Nature Portfolio guidelines for submitting code & software for further information.

## Data

Policy information about availability of data

All manuscripts must include a data availability statement. This statement should provide the following information, where applicable:

- Accession codes, unique identifiers, or web links for publicly available datasets
- A description of any restrictions on data availability
- For clinical datasets or third party data, please ensure that the statement adheres to our policy

Final checkpoints of ImmunoMatch, ImmunoMatch-kappa and ImmunoMatch-lambda are available on HuggingFace at https://huggingface.co/fraternalilab/ immunomatch. Code to run ImmunoMatch to annotate sequences can be found on Google Collaboratory (https://colab.research.google.com/github/Fraternalilab/ ImmunoMatch/blob/main/Run_ImmunoMatch.ipynb). Data and code to generate figures in this manuscript are available on GitHub (https://github.com/ Fraternalilab/ImmunoMatch). A standalone Python package to apply ImmunoMatch is available on PyPI at https://pypi.org/project/ImmunoMatch/.

## Research involving human participants, their data, or biological material

Policy information about studies with human participants or human data. See also policy information about sex, gender (identity/presentation), and sexual orientation and race, ethnicity and racism.

| | |
|---|---|
| Reporting on sex and gender | N/A. No newly generated data is presented in this study. We use publicly available datasets to validate our computational method (see Software and Code) |
| Reporting on race, ethnicity, or other socially relevant groupings | N/A. No newly generated data is presented in this study. We use publicly available datasets to validate our computational method (see Software and Code) |
| Population characteristics | N/A. No newly generated data is presented in this study. We use publicly available datasets to validate our computational method (see Software and Code) |
| Recruitment | N/A. No newly generated data is presented in this study. We use publicly available datasets to validate our computational method (see Software and Code) |
| Ethics oversight | N/A. We did not generate new data from human samples for this study.. |

Note that full information on the approval of the study protocol must also be provided in the manuscript.

# Field-specific reporting

Please select the one below that is the best fit for your research. If you are not sure, read the appropriate sections before making your selection.

☒ Life sciences  ☐ Behavioural & social sciences  ☐ Ecological, evolutionary & environmental sciences

For a reference copy of the document with all sections, see nature.com/documents/nr-reporting-summary-flat.pdf

# Life sciences study design

All studies must disclose on these points even when the disclosure is negative.

| | |
|---|---|
| Sample size | We use all sequences available to us from high-quality paired antibody repertoire datasets. We clustered sequences prior to defining training and test datasets to avoid skewing of model learning towards large expanded B cell clonotypes. |
| Data exclusions | N/A. All collected datasets were used in model training and validation following standard machine learning practices to ensure model generalisability on external datasets. |
| Replication | Whenever possible k-fold cross validation procedure was employed to demonstrate model consistency on different data subsets. The prediction model was validated on separate datasets representing a variety of biological contexts. |
| Randomization | Sequences were randomly divided into training and test sets following standard machine learning practices. We clustered CDR sequences using the CD-HIT (Fu et al. Bioinformatics 2012) algorithm prior to defining the training and test sets to exclude the chance of similar sequences appearing in both datasets and therefore minimise data leakage. These details are included in the Methods section. Randomisation in the validation datasets is not applicable. |
| Blinding | Blinding is not relevant to this study since our main aim is not to claim differences between defined groups of observations. |

# Reporting for specific materials, systems and methods

We require information from authors about some types of materials, experimental systems and methods used in many studies. Here, indicate whether each material, system or method listed is relevant to your study. If you are not sure if a list item applies to your research, read the appropriate section before selecting a response.

## Materials & experimental systems

| n/a | Involved in the study |
|-----|----------------------|
| ☒ ☐ | Antibodies |
| ☒ ☐ | Eukaryotic cell lines |
| ☒ ☐ | Palaeontology and archaeology |
| ☒ ☐ | Animals and other organisms |
| ☒ ☐ | Clinical data |
| ☒ ☐ | Dual use research of concern |
| ☒ ☐ | Plants |

## Methods

| n/a | Involved in the study |
|-----|----------------------|
| ☒ ☐ | ChIP-seq |
| ☒ ☐ | Flow cytometry |
| ☒ ☐ | MRI-based neuroimaging |

## Plants

| Seed stocks | N/A |
|-------------|-----|
| Novel plant genotypes | N/A |
| Authentication | N/A |

