## [Peer Review File · Nature Methods]

ImmunoMatch learns and predicts cognate pairing of heavy and light immunoglobulin chains

Corresponding Author: Dr Joseph Ng

Version 0:

Decision Letter:

15th Apr 2025

Dear Joseph,

Your Article, "ImmunoMatch learns and predicts cognate pairing of heavy and light immunoglobulin chains", has now been seen by 2 reviewers. As you will see from their comments below, although the reviewers find your work of considerable potential interest, they have raised a number of concerns. We are interested in the possibility of publishing your paper in Nature Methods, but would like to consider your response to these concerns before we reach a final decision on publication.

We therefore invite you to revise your manuscript to address these concerns. We strongly suggest showing applications to demonstrate how ImmunoMatch may enable basic biological studies and also show the generalizability of the method. Analyzing patient data and therapeutic antibodies is optional as it is beyond the scope of Nature Methods, so I will leave it to you.

We hope you understand that until we have read the revised paper in its entirety we cannot promise that it will be sent back for peer-review.

Link Redacted

We hope to receive your revised paper within six months. If you cannot send it within this time, please let us know. In this event, we will still be happy to reconsider your paper at a later date so long as nothing similar has been accepted for

publication at Nature Methods or published elsewhere.

OPEN SCIENCE REQUIREMENTS

REPORTING SUMMARY AND EDITORIAL POLICY CHECKLISTS

EXTENDED DATA FIGURES

DATA AVAILABILITY

All novel DNA and RNA sequencing data, protein sequences, genetic polymorphisms, linked genotype and phenotype data, gene expression data, macromolecular structures, and proteomics data must be deposited in a publicly accessible database, and accession codes and associated hyperlinks must be provided in the "Data Availability" section.

CODE AVAILABILITY

Please include a "Code Availability" subsection in the Online Methods which details how your custom code is made available. Only in rare cases (where code is not central to the main conclusions of the paper) is the statement "available upon request" allowed (and reasons should be specified).

For more information on our code sharing policy and requirements, please see:
<https://www.nature.com/nature-research/editorial-policies/reporting-standards#availability-of-computer-code>

SUPPLEMENTARY PROTOCOL

To help facilitate reproducibility and uptake of your method, we ask you to prepare a step-by-step Supplementary Protocol for the method described in this paper. We [encourage authors to share their step-by-step experimental protocols](https://www.nature.com/nature-research/editorial-policies/reporting-standards#protocols) on a protocol sharing platform of their choice and report the protocol DOI in the reference list. Nature Portfolio's protocols.io is a free-to-use and open resource for protocols; protocols deposited onto protocols.io are citable and can be linked from the published article. More details can be found at [protocols.io](https://www.protocols.io/help/publish-articles).

ORCID

Sincerely,
Madhura

Madhura Mukhopadhyay, PhD
Senior Editor
Nature Methods

Reviewers' Comments:

Reviewer #1 (Remarks to the Author):

Summary

The manuscript presents a novel machine learning approach to pair B cell receptor (BCR) heavy and light antibody chain sequences. Their model, ImmunoMatch, is trained on paired H-L antibody sequences and fine-tuned on AntiBERTa2 to distinguish between cognate H-L pairs and randomly paired sequences and represents a natural step towards solving a previously unaddressed problem. ImmunoMatch is well-validated through its application to a number of external datasets, and the pairing score's relationship to cell-type and mutation from the germline is explored and fits with current understandings of B cell development. With the exception of a lack of clarity in a few figures, the manuscript is clear and effectively written. Despite its strengths, I have concerns regarding the method's interpretability and applications, as well as other concerns regarding the model and clarity in a few places. Please see the detailed comments below.

Major Comments

1. While the recent popularity of paired single cell BCR H and L chain data makes developing a model like ImmunoMatch an obvious application of machine learning, I struggled to think of a specific use case for this model. The authors mention potential applications, but these are not sufficiently detailed to make them usable.
 - a. First, the authors suggest that researchers can use the ImmunoMatch pairing score to perform QC on single-cell BCR datasets. However, it isn't clear what this would actually look like. Are there cases where we'd expect single cell data sets to have unlikely H and L pairs that result from technical artifacts? How would ImmunoMatch distinguish those from true H and L pairs? What distribution of the pairing scores would indicate good or bad single-cell BCR data? Doing so and providing more guidance would be necessary to allow ImmunoMatch to help with QC. Even then, this doesn't seem like a highly impactful application.
 - b. Second, the pairing score could potentially be used as a scoring system for antibody pairing. However, single cell sequencing data with paired H and L chains is widely available, if expensive. Thus, the need for computationally inferring H/L pairs is reduced. Thus, it's not clear what setup this application is really targeting. Perhaps a scenario where bulk BCR sequencing is performed on H and L chains separately and then pairings are computationally inferred? If so, it's not clear how or how well that would work. I would be concerned about the frequency of false positives on a large dataset scale.
 - c. Third, the authors suggest that the ImmunoMatch pairing score can be used in therapeutic antibody developability. While I think it could be used in this context, the authors do not present the distribution of pairing scores on the set of therapeutic antibodies they consider, which I believe is needed. If the pairing score is above some threshold for all (or nearly all) of the therapeutics, it would make sense to use a low pairing score to filter out potential antibodies.

2. It is unclear how the ImmunoMatch pairing score should be interpreted. It is not a probability of a heavy and light chain pairing, but the authors do not explain much about what the score means. What is the distribution of the pairing score on the test datasets? Should it be interpreted on an absolute scale or a relative scale? To elaborate on this, imagine we have one heavy chain with five potential light chains. The paired chains are run through the ImmunoMatch model, and the pairing scores are 0.01, 0.01, 0.02, 0.05, 0.001. The fourth pair has the highest pairing score, but how confident should we be that this represents a functional pairing?

3. Regarding antibody developability and Figure 4, as alluded to above, I believe a more compelling case for including ImmunoMatch in the development process would be to show that the pairing scores for real therapeutic antibodies are significantly higher than the randomly generated pairs. At the very least, I think it is necessary to present the distribution of pairing scores on the real therapeutic antibodies. If the pairing score for developed antibodies is not typically high, or at least higher than random heavy + light chain pairs, I struggle to understand how ImmunoMatch would fit in to the development process.

4. It is understandable why the pseudo-negative paired chains used to train the model are constructed by randomly pairing. However, I worry that if some real H/L pairings are likely to occur by random chance (for example, if both chains are very common), they would frequently appear in the pseudo-negative data. I think it would be worth quantifying how frequently H/L pairs in the pseudo-negative controls appear in real single-cell BCR datasets, particularly in the disease contexts that ImmunoMatch is being applied to. This would help filter out false negatives and potentially improve the performance of the model without much additional work.

Minor Comments

1. The training data is drawn from only 6 donors. It appears that all 6 were currently healthy individuals, which may be what drives the low sample size. However, if this is the case, ImmunoMatch is likely biased towards accurately pairing heavy and light chains in healthy individuals. How well does ImmunoMatch perform when analyzing patients in different contexts, for instance infections/vaccination, or autoimmune disease? This is important as it would likely be applied in these cases.

2. Generally, models trained on less diverse datasets perform better in similar contexts, and models trained on more general datasets are more generalizable. However, the authors state that while lambda light chains are more similar than kappa light chains, the H-L lambda pairing model is more generalizable. I follow the argument that the lambda light chains' "rescue role" may allow the H-L lambda model to learn rules for pairing H-L kappa chains, but I still found it surprisingly that a model trained on a less diverse dataset would perform better. I believe this argument should be elaborated on to be more convincing.

3. The authors state "A number of computational methods aim to simulate antibody repertoires for benchmarking applications, or for learning the statistics of germline gene usage repertoire data [102, 103, 104]. Our results suggest that simulation without constraining paired H-L gene usage could significantly oversample dysfunctional sequences incapable of contributing to stable H-L pairs, which would call into question the reliability of these approaches generating useful benchmarking single-cell, paired-chain datasets resembling real-life antibody repertoires." I follow this logic, and agree that this is a drawback of existing approaches that randomly pair H and L chains. However because ImmunoMatch is not a generative model, it isn't clear how it could be used to simulate new data.

4. In Figure 3, the authors show that, generally, the pairing score seems to increase as BCRs undergo affinity maturation. It would be beneficial to see a scatter plot of SHM vs. pairing score, annotated for cell type. Statistically testing this relationship while adjusting for cell type would add further support.

5. Figure 3d, showing the ImmunoMatch pairing scores for BCRs from leukemia and lymphoma, is very interesting and a well-made figure. However, the sample size (123 total paired sequences) is very low, which detracts from its strength. Are there any further datasets the authors could include in this analysis to increase the sample size in terms of sequences tested?

Reviewer #1 (Remarks on code availability):

Code was installable and usable.

Reviewer #2 (Remarks to the Author):

A. Summary of the key results

The authors present a novel machine learning framework ImmunoMatch which can differentiate between randomly assigned heavy chain light chain pairs and actual cognate pairs. ImmunoMatch is based on an antibody specific language model trained on single cell B cell sequencing data. They show that greater accuracy is reached when the models are made separately for each light chain type, kappa and lambda.

B. Originality and significance: if not novel, please include reference

While other research has been done on using machine learning tools and AI to identify repertoires and immune response no previous application has been created to identify light chain / heavy chain interactions. This is a significant advance that can have impact both for drug design and as a tool to study how different positions at the heavy chain / light chain interface impact

heavy chain light chain association

C. Data & methodology: validity of approach, quality of data, quality of presentation

The data and methodology seem valid and correct.

D. Appropriate use of statistics and treatment of uncertainties

The statistical inferences underlying the creation of ImmunoMatch are clearly described as is the randomization used to create control models and their comparison to the observations of the trained ImmunoMatch. It is not completely clear to what extent sufficient data was used for the training and that the addition of data or the choice of other repertoires would change the characteristics of the model. all in all the training is based on 6 individuals (allbeit with several million heavy chain/ light chain pairs).

E. Conclusions: robustness, validity, reliability

Other than the comment above the conclusions are well described and appear valid

F. Suggested improvements: experiments, data for possible revision

In line 39 and 40 the authors state: "Secondly, mutations accumulated in the antibody variable region exponentiate repertoire diversity"

What does the word "exponentiate" mean? Did the authors mean to write something else?

In line 181 you state: "This suggests that ImmunoMatch- λ is more generalisable in learning pairing rules for antibodies with κ and λ light chains.

Does accuracy only measure frequency of true possessives and lack of false negatives or also tell us how often true positives are missed. i.e. is the lambda model of ImmunoMatch only avoiding bad kappa/ heavy chain pairs with out also missing some good pairs (That the kappa model would catch) or does one come at the expense of the other.

In line 231 the authors write – "both health and disease" They should write "both healthy and diseased".

G. References: appropriate credit to previous work?

yes

H. Clarity and context: lucidity of abstract/summary, appropriateness of abstract, introduction and conclusions

The paper is clearly written

Reviewer #2 (Remarks on code availability):

The code is well annotated and includes a usefull google colab tool for ease of interaction with the code. It is functioning and performs as described in the paper.

Version 1:

Decision Letter:

Our ref: NMETH-A59849A

4th Sep 2025

Dear Joseph,

Thank you for submitting your revised manuscript "ImmunoMatch learns and predicts cognate pairing of heavy and light immunoglobulin chains" (NMETH-A59849A). It has now been seen by the original referees and their comments are below. The reviewers find that the paper has improved in revision, and therefore we'll be happy in principle to publish it in Nature Methods, pending minor revisions to satisfy the referees' final requests and to comply with our editorial and formatting guidelines.

We are now performing detailed checks on your paper and will send you a checklist detailing our editorial and formatting requirements within two weeks or so. Please do not upload the final materials and make any revisions until you receive this additional information from us. Please address the remaining reviewer comments and send us a point-by-point rebuttal when resubmitting.

TRANSPARENT PEER REVIEW

Please note: we allow redactions to authors' rebuttal and reviewer comments in the interest of confidentiality. If you are concerned about the release of confidential data, please let us know specifically what information you would like to have removed. Please note that we cannot incorporate redactions for any other reasons. Reviewer names will be published in the peer review files if the reviewer signed the comments to authors, or if reviewers explicitly agree to release their name. For more information, please refer to our [FAQ](https://www.nature.com/documents/nr-transparent-peer-review.pdf)

page.

ORCID

Sincerely,
Madhura

Madhura Mukhopadhyay, PhD
Senior Editor
Nature Methods

Reviewer #1 (Remarks to the Author):

The authors have done a great job of addressing our prior critiques. We believe this revised manuscript will be highly impactful.

Kenneth Hoehn and Hunter Melton

Reviewer #1 (Remarks on code availability):

Code is installable and usable

Reviewer #2 (Remarks to the Author):

Given that my comments in the last review were quite focused as was the authors response I will in this second review only discuss these points.

I commented: "It is not completely clear to what extent sufficient data was used for the training and that the addition of data or the choice of other repertoires would change the characteristics of the model. "

To which the authors responded by adding 5 individuals to the original 6 used to create their tool. The authors note that this actually degraded their accuracy by generating more false positive classifications (i.e. identifying heavy/light chain pairs as associating when they do not. This result does not alleviate my doubts on sampling, quite the contrary. it would have been useful if, rather than just adding a comparison showing that using only one individual breaks things even more, the authors tried to check what would be the accuracy if different set of six individuals were used 6 and up that make up these 11 individuals (i.e. calculating accuracy for all sub groups of they tried to characterize why the data is degrading the results. Given the novelty of this study it is still important to publish this result but with the caveat of the limitations of existing data levels to creating consistently accurate models.

It is unclear why the authors do not in any way discuss in the corrected paper their extended model that they presented to the reviewers. Moreover, why in the reviewer response they only use the external test while in the paper they mostly use the withheld test. The authors should contend with their extended test results. Testing to what extent different sets of individuals change the test's accuracy in different ways or at the very least explain why the first six chosen are so efficacious for their model.

Reviewer #2 (Remarks on code availability):

I did not reassess the code as the authors response did not indicate any change in the code from my last review

Reviewer #4 (Remarks to the Author):

I co-reviewed this manuscript with one of the reviewers who provided the listed reports. This is part of the Nature Methods initiative to facilitate training in peer review and to provide appropriate recognition for Early Career Researchers who co-review manuscripts.

Reviewer #4 (Remarks on code availability):

Code was installable and usable. Code to reproduce figures and README were provided.

Version 2:

Decision Letter:

17th Oct 2025

Dear Joseph,

I am pleased to inform you that your Article, "ImmunoMatch learns and predicts cognate pairing of heavy and light immunoglobulin chains", has now been accepted for publication in Nature Methods. The received and accepted dates will be Feb 16, 2025 and Oct 17, 2025. This note is intended to let you know what to expect from us over the next month or so, and to let you know where to address any further questions.

Over the next few weeks, your paper will be copyedited to ensure that it conforms to Nature Methods style. Once your paper is typeset, you will receive an email with a link to choose the appropriate publishing options for your paper and our Author Services team will be in touch regarding any additional information that may be required. It is extremely important that you let us know now whether you will be difficult to contact over the next month. If this is the case, we ask that you send us the contact information (email, phone and fax) of someone who will be able to check the proofs and deal with any last-minute problems.

Authors may need to take specific actions to achieve compliance with funder and institutional open access mandates.

If your research is supported by a funder that requires immediate open access (e.g. according to [Plan S principles](https://www.springernature.com/gp/open-science/plan-s-compliance) or the [NIH public access policy](https://www.springernature.com/gp/open-science/us-federal-agency-compliance)) then you should select the gold OA route, and we will direct you to the compliant route where possible. Because authors warrant under our subscription licensing terms that they haven't committed to licensing any version of their article under a licence inconsistent with the terms of our agreement – including the applicable embargo period – publication under the subscription model isn't suitable for authors whose funders require no embargo.

If you are active on Twitter/X or Bluesky, please e-mail me your and your coauthors' handles so that we may tag you when the paper is published.

Best regards,
Madhura

Madhura Mukhopadhyay, PhD
Senior Editor

** Visit the Springer Nature Editorial and Publishing website at www.springernature.com/editorial-and-publishing-jobs for more information about our career opportunities. If you have any questions please click here. **
